# Co-targeting CDK4/6 and AKT with endocrine therapy prevents progression in CDK4/6 inhibitor and endocrine therapy-resistant breast cancer

Carla L. Alves [1]✉, Sidse Ehmsen [1,2,8], Mikkel G. Terp[1,8], Neil Portman[3,4], Martina Tuttolomondo [1], Odd L. Gammelgaard [1], Monique F. Hundebøl[1], Kamila Kaminska[5], Lene E. Johansen[1], Martin Bak [6], Gabriella Honeth[5], Ana Bosch[5], Elgene Lim [3,4] & Henrik J. Ditzel [1,2,7]✉

CDK4/6 inhibitors (CDK4/6i) combined with endocrine therapy have shown impressive efficacy in estrogen receptor-positive advanced breast cancer. However, most patients will eventually experience disease progression on this combination, underscoring the need for effective subsequent treatments or better initial therapies. Here, we show that triple inhibition with fulvestrant, CDK4/6i and AKT inhibitor (AKTi) durably impairs growth of breast cancer cells, prevents progression and reduces metastasis of tumor xenografts resistant to CDK4/6i-fulvestrant combination or fulvestrant alone. Importantly, switching from combined fulvestrant and CDK4/6i upon resistance to dual combination with AKTi and fulvestrant does not prevent tumor progression. Furthermore, triple combination with AKTi significantly inhibits growth of patient-derived xenografts resistant to combined CDK4/6i and fulvestrant. Finally, high phospho-AKT levels in metastasis of breast cancer patients treated with a combination of CDK4/6i and endocrine therapy correlates with shorter progression-free survival. Our findings support the clinical development of ER, CDK4/6 and AKT co-targeting strategies following progression on CDK4/6i and endocrine therapy combination, and in tumors exhibiting high phospho-AKT levels, which are associated with worse clinical outcome.

[1] Department of Cancer and Inflammation Research, Institute of Molecular Medicine, University of Southern Denmark, Odense, Denmark. [2] Department of Oncology, Institute of Clinical Research, Odense University Hospital, Odense, Denmark. [3] Garvan Institute of Medical Research, Sydney, NSW, Australia. [4] St. Vincent's Clinical School, Faculty of Medicine, University of New South Wales Sydney, Sydney, NSW, Australia. [5] Division of Oncology and Pathology, Department of Clinical Sciences, Lund University, Lund, Sweden. [6] Department of Pathology, Sydvestjysk Sygehus, Esbjerg, Denmark. [7] Academy of Geriatric Cancer Research (AgeCare), Odense University Hospital, Odense, Denmark. [8] These authors contributed equally: Sidse Ehmsen, Mikkel G. Terp.
✉email: calves@health.sdu.dk; hditzel@health.sdu.dk

Endocrine therapy comprises the most effective targeted therapies for the treatment of estrogen receptor-positive (ER+) breast cancer. However, the development of resistance to these agents remains a major clinical challenge. The role of cyclin D-CDK4/6 signaling in ER+ breast cancer tumorigenesis and endocrine resistance is well described[1–5]. Importantly, studies have shown that ER+ breast cancer resistance to endocrine therapy is dependent on the cyclin D-CDK4/6 pathway[6,7]. Together, these data supported the clinical investigation of several CDK4/6 inhibitors (CDK4/6i), including palbociclib, ribociclib, and abemaciclib, in combination with endocrine therapy. Clinical studies have demonstrated that combined CDK4/6i and endocrine therapy significantly improves progression-free survival (PFS) and overall survival (OS) compared to endocrine therapy alone in ER+ advanced breast cancer, resulting in the approval of CDK4/6i in the first-line setting combined with an aromatase inhibitor (AI). In addition, the combination of CDK4/6i and ER degrader fulvestrant was approved for use following progression on initial AI monotherapy[8–13].

The link between deregulated PI3K/AKT-mTOR pathway and endocrine resistance is also well known[14]. Everolimus, a mTORC1 inhibitor, was shown to prolong PFS in combination with the AI exemestane in ER+ advanced breast cancer after progression on non-steroidal AI[15]. However, inhibition of mTORC1 induces a negative feedback loop that activates AKT, limiting the efficacy of mTORC1 inhibitors[16]. In addition, alpelisib, an alpha-specific PI3K inhibitor (PI3Ki), has recently been approved for the treatment of *PIK3CA*-mutated ER+ advanced breast cancer that progressed on previous endocrine therapy[17]. Notably, it has been suggested that direct blockade of AKT may provide a better treatment option for endocrine-resistant breast cancer, affecting cell survival and ER ligand-independent signaling in both *PIK3CA*-mutant and wild-type tumors[18]. Currently, there are several AKT inhibitors (AKTi) under clinical investigation, including the pan-AKT kinase catalytic inhibitor capivasertib (AZD5363) that, in combination with fulvestrant, has recently been demonstrated to improve PFS in ER+ metastatic breast cancer patients whose tumors progressed on an AI[19]. Interestingly, no difference in response was observed between tumors with PI3K/PTEN/AKT pathway activation due to genetic alterations in PI3K/PTEN/AKT and tumors without such genetic alterations. A phase III trial is currently evaluating capivasertib in combination with fulvestrant in ER+ metastatic breast cancer patients following progression on an AI (CAPItello-291). Furthermore, a current phase Ib/III trial is evaluating AKTi capivasertib plus palbociclib and fulvestrant vs. palbociclib and fulvestrant in ER+ locally advanced, unresectable, or metastatic breast cancer (CAPItello-292).

Nonetheless, some patients have tumors that do not respond to CDK4/6i, and a significant number of patients with tumors that initially respond to these drugs will progress, underscoring the need to identify biomarkers and develop more rational drug combinations. Recently, CDK4/6i was found to sensitize *PIK3CA*-mutant tumors to PI3Ki and, conversely, mTORC1/2 inhibitors inhibited the growth of CDK4/6i-resistant cells[20,21]. Furthermore, activation of the PI3K/AKT-mTOR pathway has been shown to be a mechanism of early adaptive resistance to CDK4/6i[22]. These data support the use of therapeutic strategies targeting both pathways to prevent the compensatory pathway activation involved in drug resistance. However, combining the available agents targeting the PI3K/AKT-mTOR pathway with the approved CDK4/6i and ER-targeted therapies results in many possible combinations in different lines of therapy, which complicates determining the optimal therapeutic strategy for individual patients. Moreover, overlapping toxicity and high costs further complicate the addition of targeted agents to standard treatments[23].

Here, we demonstrate that a triple combination of the AKTi, CDK4/6i, and fulvestrant is required to durably impair growth and prevent progression in ER+ breast cancer cell lines and tumor xenografts resistant to combined therapy with fulvestrant and CDK4/6i or fulvestrant alone. Furthermore, the triple combination significantly inhibited the growth of patient-derived xenografts (PDXs) resistant to combined CDK4/6i and fulvestrant. Importantly, switching from fulvestrant and CDK4/6i combination, upon resistance, to the combination of AKTi and fulvestrant did not prevent tumor progression. These data suggest that triple combination with AKTi, CDK4/6i, and fulvestrant represents a therapeutic option for tumors that will relapse on standard therapy with fulvestrant alone or in combination with CDK4/6i. Further, we found that high levels of phospho-AKT (p-AKT) in metastatic lesions from ER+ breast cancer patients treated with combined endocrine therapy and CDK4/6i in the advanced setting correlated with shorter PFS. Our findings support the clinical development of triple combinations with fulvestrant, CDK4/6i, and AKTi in pre-treated ER+ advanced breast cancer, particularly in tumors exhibiting high levels of p-AKT, to improve patient survival.

## Results

**Fulvestrant, CDK4/6i and AKTi triple combination therapy is required for durable growth inhibition of fulvestrant-resistant breast cancer cells**. We previously demonstrated that cooperation between CDK6 and AKT confers resistance to fulvestrant in ER+ breast cancer cell lines[24]. Furthermore, recent studies have shown that inhibitors of the PI3K/AKT-mTOR pathway synergize with agents targeting the cyclin D/CDK4-6/Rb axis, which supports the use of combinational strategies with inhibitors of both pathways[21,25,26]. However, which therapeutic strategy, endocrine therapy combined with either a CDK4/6 or a PI3K/AKT-mTOR inhibitor, sequential treatments with these combinations or upfront triple combination, is the best as first-line treatment in endocrine-resistant and endocrine-sensitive tumors remains to be defined. Herein, we assessed the efficacy of the AKTi capivasertib, CDK4/6i palbociclib, and the ER degrader fulvestrant as single agents and in double and triple combinations in ER+MCF-7, T47D, and ZR-75-1 fulvestrant-resistant and fulvestrant-sensitive breast cancer cell models. The concentrations of CDK4/6i and AKTi used were determined based on the highest $IC_{50}$ between parental and resistant cell lines of each cell model (Supplementary Fig. S1). As expected, fulvestrant alone induced a marked decrease in the growth of fulvestrant-sensitive cells (Fig. 1A, E, and I). Although combined fulvestrant and CDK4/6i almost completely inhibited the growth of all the fulvestrant-sensitive cell lines and fulvestrant-resistant ZR-75-1 R cells, it had only a limited effect on fulvestrant-resistant 182R-1 (MCF-7 based) and T47D R cells (Fig. 1A–B, E–F, and I–J). Moreover, a triple combination with fulvestrant, CDK4/6i, and AKTi decreased growth to a greater degree than observed with fulvestrant and CDK4/6i, and was needed to maintain growth inhibition of fulvestrant-resistant 182R-1 and T47D R cells (Fig. 1A–B and E–F). In line with these findings, we observed that the triple combination more potently impaired the viability of all cell lines compared to the fulvestrant and CDK4/6i combination (Fig. 1C–D, G–H, and K–L. The efficacy of combined fulvestrant and AKTi was comparable to the approved fulvestrant and CDK4/6i combination (Fig. 1A–L). The additional growth inhibitory effect of the triple combination compared to the standard fulvestrant and CDK4/6i combination was, at least in part, a result of induction of apoptosis (Fig. 2A–C) and cleaved-PARP levels (Fig. 2D). Calculations of the combination index (CI) showed that fulvestrant, CDK4/6i and AKTi exhibited synergistic

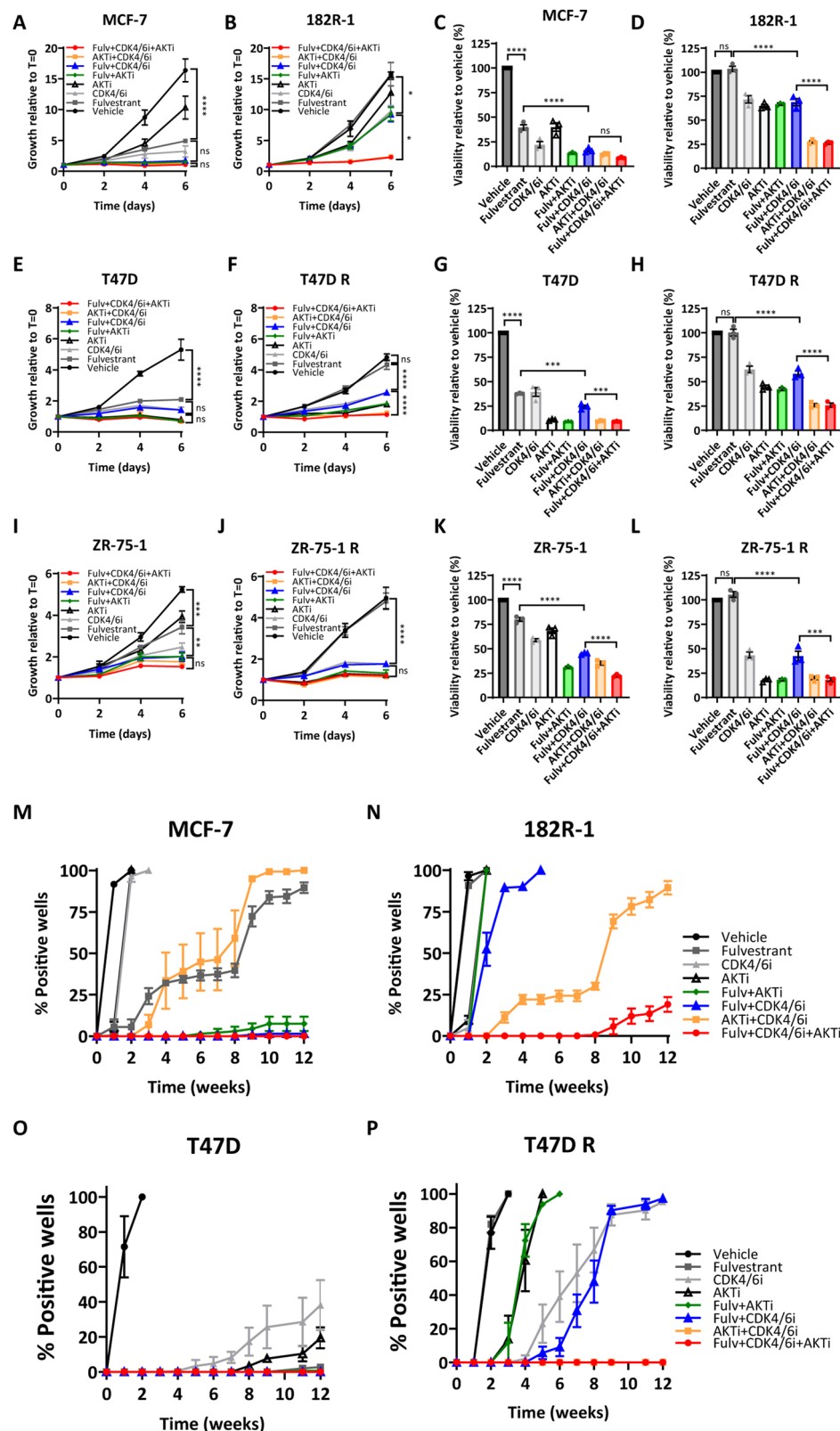

activity when combined, except for combined fulvestrant and AKTi in ZR-75-1 R cells (Supplementary Fig. S2). Finally, we examined whether the triple combination could delay the emergence of resistance in a colony outgrowth assay (Fig. 1M–P and Supplementary Fig. S3). We observed that the triple combination suppressed the growth of 182R-1 cells up to 8 weeks and T47D R over the entire 12 weeks of treatment (Fig. 1N and P), while

relatively rapid outgrowth was observed with combined fulvestrant and CDK4/6i or AKTi (1 week for 182R-1 and 3–5 weeks for T47D R). In contrast, no or very slow outgrowth was observed in parental MCF-7 and T47D cells treated with the combination of fulvestrant with either CDK4/6i or AKTi or the triple combination (Fig. 1M and O). ZR-75-1 R cells showed high sensitivity to AKTi and CDK4/6i single agents in the long-term growth assay

**Fig. 1 Combined fulvestrant, CDK4/6i and AKTi is required to significantly and durably inhibit growth in ER+ fulvestrant-resistant breast cancer cell lines.** The effect of fulvestrant (Fulv, 100 nM), CDK4/6 inhibitor (CDK4/6i, palbociclib, 200 nM in the MCF-7 and T47D cell models; 5 μM in the ZR-75-1 cell model) and AKT inhibitor (AKTi, capivasertib, 500 nM in the MCF-7 cell model; 200 nM in the T47D model; 150 nM in the ZR-75-1 cell model), as single agents or in double and triple combinations, was assessed in all cell lines by crystal violet growth assay (**A**, **B**, **E**, **F**, **I** and **J**) and CellTiter-Blue viability assay (**C**, **D**, **G**, **H**, **K**, and **L**) performed over 6 days. Outgrowth of resistant colonies was investigated in MCF-7 (**M** and **N**) and T47D models (**O** and **P**) by weekly evaluation of the percentage of 48 wells at 50% or greater confluence (positive wells) over 12 weeks. Experiments were conducted in three biological replicates and data are shown as mean ± SEM. Asterisks indicate significant differences in the one-way ANOVA test at day 6 (*$0.01 < p < 0.05$, **$0.001 < p < 0.01$, ***$0.0001 < p < 0.001$, and ****$p < 0.0001$).

(Supplementary Fig. S3B). Importantly, the concentration of CDK4/6i used in ZR-75-1 cells based on the IC$_{50}$ value was considerably high (Supplementary Fig. S1A) and palbociclib has been shown to inhibit the proteasomal regulator DYRK1A at 2 μM[27], suggesting an unspecific effect of palbociclib on the growth of ZR-75-1 cells. Nevertheless, our data suggest that the approved treatment with fulvestrant combined with a CDK4/6i efficaciously inhibits the growth of endocrine-sensitive and some endocrine-resistant breast cancer cells, but simultaneous inhibition of ER, CDK4/6, and AKT is required to durably suppress the growth of most the endocrine-resistant cells of different breast cancer models.

**Combined targeting of CDK4/6 and AKT efficiently inhibits cyclin D/CDK4-6/Rb and PI3K/AKT-mTOR pathways in ER+ breast cancer cell lines.** Next, we evaluated the expression and activation of key molecules of the three cellular signal pathways (ER, cyclin D/CDK4-6/Rb, and PI3K/AKT-mTOR) 72 h after treatment with the targeted inhibitors capivasertib, palbociclib, and fulvestrant in MCF-7, T47D, and ZR-75-1 breast cancer cell line models (Fig. 2E), as this is when immediate growth suppression induced by palbociclib is released[22]. We observed that the triple combination led to a marked ER decrease and significantly reduced phosphorylation levels of Rb, PRAS40, and S6 proteins in both fulvestrant-sensitive and -resistant cells (Fig. 2E), indicating inhibition of ER, cyclin D/CDK4-6/Rb, and PI3K/AKT-mTOR pathways. Treatment with AKTi induced phosphorylation of AKT, which maintains the protein in a hyper-phosphorylated, catalytically inactive state, as previously described[18]. Notably, none of the single agents or combinations without simultaneous targeting of AKT and CDK4/6 produced similarly profound inhibition of the downstream targets of the three pathways compared to the triple combination, particularly in fulvestrant-resistant cells (Fig. 2E). Interestingly, we observed that 182R-1 and ZR-75-1 R fulvestrant-resistant cells exhibited higher levels of p-AKT compared to their corresponding parental cell lines, while T47D R cells showed slightly lower expression of p-AKT compared to parental cells (Fig. 2F). No significant changes were observed in total AKT levels in resistant vs. sensitive cells in the three fulvestrant-resistant cell models (Fig. 2F). Noteworthy, 182R-1 is *PIK3CA* mutant, while ZR-75-1 R is *PIK3CA* wild-type, suggesting that AKT activation is not associated with *PIK3CA* mutation status. Also, MCF-7 cells showed the lowest p-AKT S473 across all models (Fig. 2F) and were treated with the highest dose of AKTi (500 nM), while ZR-75-1 and T47D cells exhibited higher p-AKT S473 and were treated with lower doses of AKTi (150 nM and 200 nM, respectively), which suggests that high p-AKT S473 levels correlate with higher AKTi sensitivity, as previously shown[28]. The different expression pattern of p-AKT in T47D/T47D R cells might be associated with its remarkably high expression level in T47D cells compared to the other sensitive cell lines. Nevertheless, these observations suggested that p-AKT could be a potential marker for the identification of fulvestrant-resistant tumors that are likely to benefit from treatment with the triple combination.

**Co-targeting CDK4/6 and AKT prevents progression of ER+ breast xenografts resistant to fulvestrant.** Next, we evaluated the efficacy of fulvestrant (100 mg/kg), CDK4/6i (50 mg/Kg), and AKTi (100 mg/kg) as monotherapies and in different combinations in vivo, using mice bearing 182R-1 tumors. Both CDK4/6i and AKTi were administered orally 5 days a week whereas fulvestrant was administered subcutaneously once a week. Fulvestrant alone induced tumor regression in MCF-7 xenografts (Fig. 3A and C) and, as expected, did not affect the growth of 182R-1 fulvestrant-resistant tumors (Fig. 3B and C). Similar results were observed when comparing the end-point tumor weight (Supplementary Fig. S4A and S4B). CDK4/6i as mono-therapy reduced tumor growth over a period of 6 weeks, but both double and triple combinations inhibited growth to a greater extent ($p = 0.02$ and $p = 0.01$, respectively) and resulted in tumor regression (Fig. 3D and Supplementary Fig. S4C). Interestingly, hematoxylin and eosin (HE) staining of 182R-1 tumors treated for 6 weeks with a vehicle, fulvestrant alone, fulvestrant + CDK4/6i, and triple combination with AKTi showed that tumors treated with vehicle or fulvestrant predominantly consisted of vital tumor cells. Tumors treated with combined CDK4/6i and fulvestrant were smaller and contained infiltrating fat cells that were more pronounced in tumors treated with the triple combination wherein only smaller tumor islets containing central degeneration surrounded by fat tissue were observed (Supplementary Fig. S4G). Neither fulvestrant nor AKTi monotherapy or combined fulvestrant and AKTi significantly inhibited tumor growth (Supplementary Fig. S4E and S4F). Although the difference between the standard combination of fulvestrant and CDK4/6i and triple combination treatment was not significant, a smaller mean of tumor volume and weight were observed for the triple combination group (Fig. 3D and Supplementary Fig. S4C). As the treatment was initiated while the tumors were relatively small (50 mm³), we next examined whether the triple combination was more effective than the standard double combination in reducing the volume of tumors allowed to expand to a larger size (250 mm³) before initiating treatment. Interestingly, significantly greater tumor regression was induced by the triple combination compared to the standard double therapy in these larger tumors at the endpoint, as evaluated by the parametric *t*-test ($p = 0.01$) and the non-parametric Wilcoxon test ($p = 0.009$, Fig. 3E and Supplementary Fig. S4D). Notably, tumor regression was most prominent during the first 3 weeks of treatment and subsequently stabilized in both the triple and double combination groups (Fig. 3E). Importantly, the triple combination completely blocked tumor regrowth during 8 weeks of treatment, while tumors treated with the standard double combination of fulvestrant and CDK4/6i started to expand after 6 weeks of treatment, suggesting outgrowth of resistant clones (Fig. 3E). We found a statistically significant difference in tumor growth rate between double and triple combinations with linear mixed-effects models (GR 7.47, $p = 0.009$, CI 1.88–13.07). Although no significant difference was observed in cleaved caspase-3, we observed decreased expression of the proliferation marker Ki67 in all fulvestrant-resistant tumors treated with the triple combination compared to

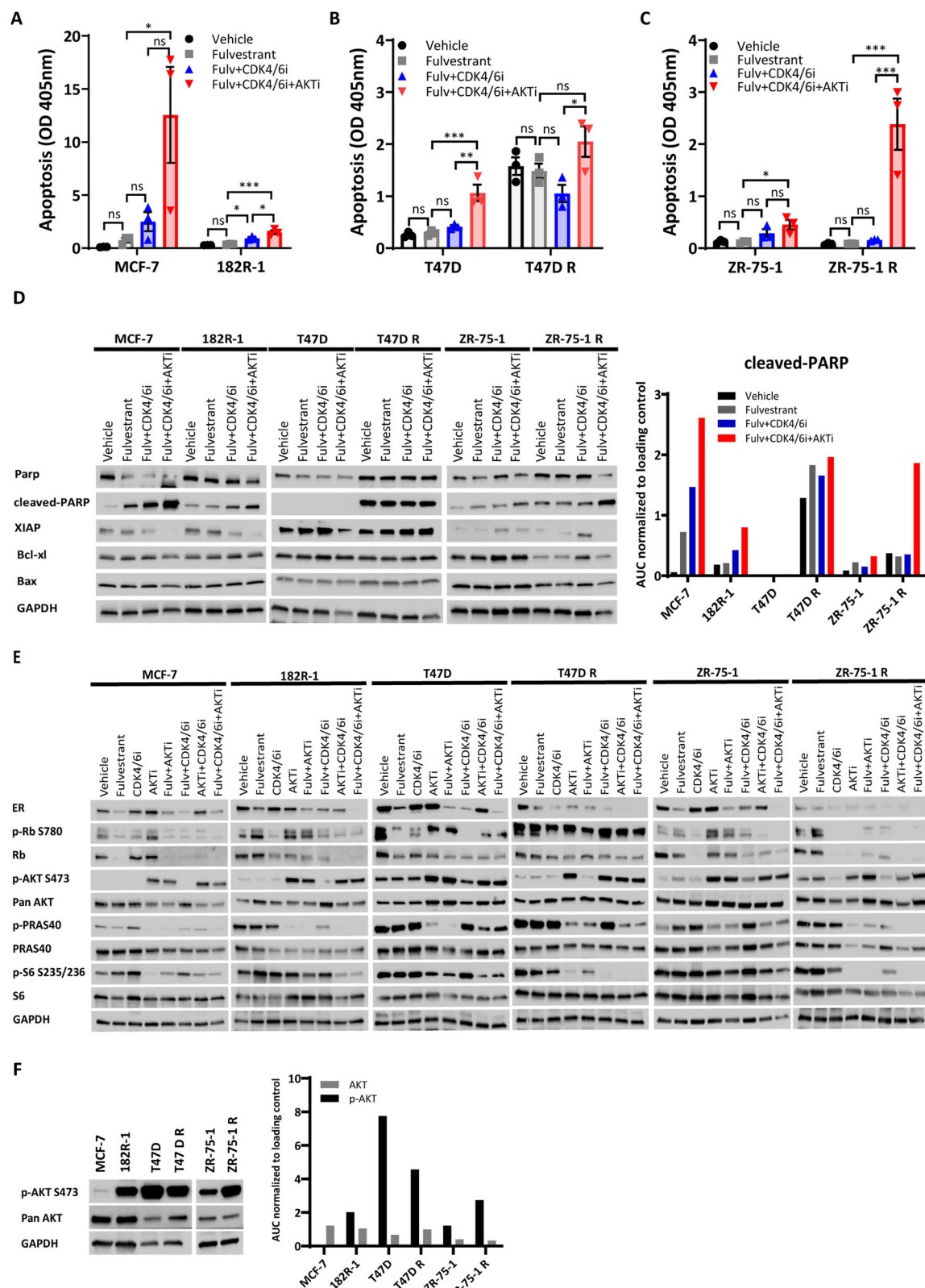

double-combination (Supplementary Fig. S4H), which is in line with our in vitro findings (Fig. 1A–D). Furthermore, HE staining showed that tumors treated with the triple combination contained large areas of degeneration and reactive fibrosis (within the indicated circles) surrounded by vital tumor tissue, while only small areas of degeneration and reactive fibrosis were observed in

tumors treated with combined CDK4/6i and fulvestrant, and the vital tumor tissue areas were much larger in these tumors (Supplementary Fig. S4H). Together, these findings support the addition of AKTi to the standard combination of fulvestrant and CDK4/6i to maintain inhibition of tumor growth in fulvestrant-resistant tumors.

**Fig. 2 Combined targeting of ER, CDK4/6 and AKT efficiently inhibits cyclin D/CDK4-6/Rb and PI3K/AKT-mTOR pathways in ER+ breast cancer cell lines.** Apoptotic levels were determined by evaluating the presence of cytoplasmic nucleosomes in MCF-7 (**A**), T47D (**B**), and ZR-75-1 (**C**) cells using an ELISA cell death detection assay. Data are shown with error bars representing mean ± SEM of three biological replicates. Asterisks indicate significant differences in the one-way ANOVA test (*$0.01 < p < 0.05$, **$0.001 < p < 0.01$, ***$0.0001 < p < 0.001$, and ****$p < 0.0001$). **D** Western blot analysis of apoptosis markers in the three fulvestrant-resistant breast cancer models. Densitometry analysis of Western blot bands of cleaved-PARP was performed using ImageJ software. Data are shown as the area under the curve (AUC) normalized to loading control. **E** Western blot analysis of key signal transduction proteins in the three fulvestrant-resistant breast cancer models. **F** Western blotting analysis of p-AKT (S473) and total AKT expression in the three fulvestrant-resistant breast cancer models. Densitometry analysis of Western blot bands of p-AKT (S473) and total AKT was performed using ImageJ software. Data are shown as the area under the curve (AUC) normalized to loading control. GAPDH was used as a loading control for Western blotting analysis. For all Western blots, a representative of two biological replicates is shown. Both apoptosis assay and harvesting protein for Western blotting analysis were performed 3 days after treatment with fulvestrant (Fulv, 100 nM), CDK4/6 inhibitor (CDK4/6i, palbociclib, 200 nM in the MCF-7 and T47D cell models; 5 µM in the ZR-75-1 cell model) and AKT inhibitor (AKTi, capivasertib, 500 nM in the MCF-7 cell model; 200 nM in the T47D model; 150 nM in the ZR-75-1 cell model).

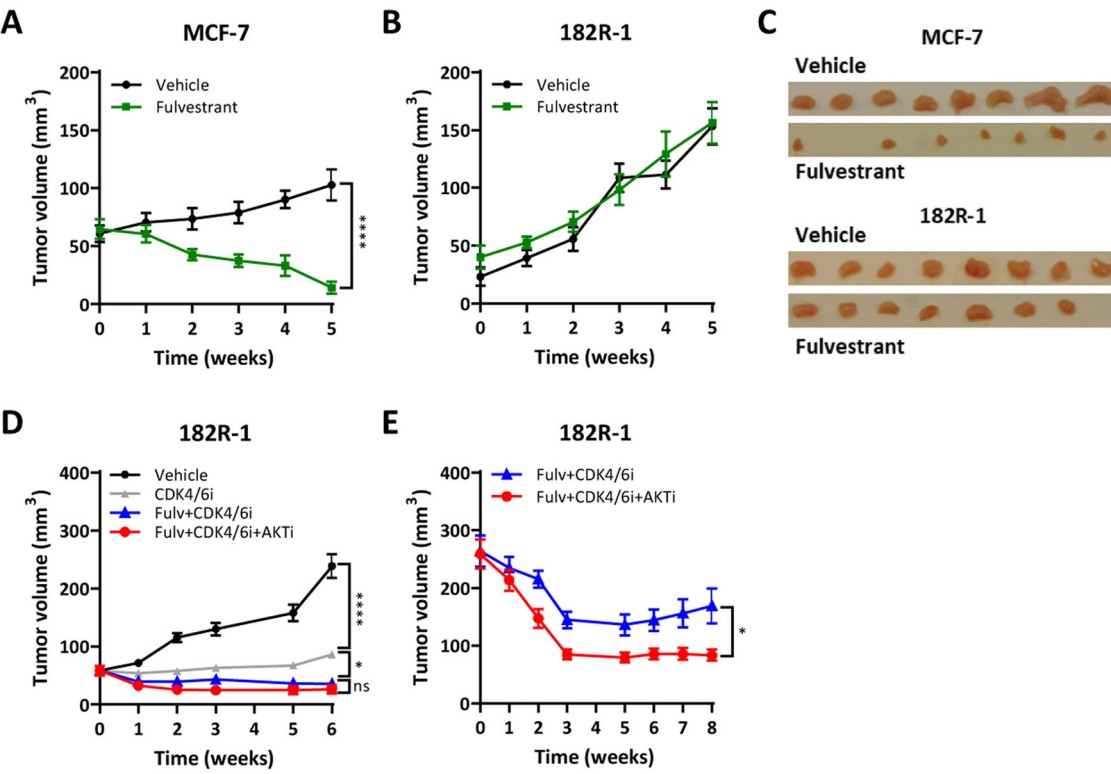

**Fig. 3 Combined inhibition of CDK4/6 and AKT prevents progression in tumor xenografts resistant to fulvestrant.** Tumor growth curves of MCF-7 (**A**) and 182R-1 (**B**) tumors following treatment with fulvestrant (Fulv, 100 mg/Kg bodyweight; $N = 7$) or vehicle (castor oil, $N = 8$) administered subcutaneously once a week. Treatment was initiated when tumors reached 50 mm³ and continued for 5 weeks. **C** Mice were sacrificed on week 5 and MCF-7 and 182R-1 tumors were excised. Tumor growth curves of 182R-1 tumors treated with CDK4/6 inhibitor (CDK4/6i, palbociclib, 50 mg/Kg bodyweight; $N = 8$) alone (**D**), in combination with fulvestrant (100 mg/Kg bodyweight; $N = 7$ and $N = 10$) (**D** and **E**), or in combination with both AKT inhibitor (AKTi, capivasertib 100 mg/Kg bodyweight) and fulvestrant (100 mg/Kg bodyweight; $N = 6$ and $N = 10$) (**D** and **E**), or vehicle (castor oil and 25% w/v HPB cyclodextrin; $N = 8$) (**D**). CDK4/6i and AKTi were administered by oral gavage once daily for 5 days a week when tumors reached 50 mm³ (**D**) or 250 mm³ (**E**), and treatment was continued for up to 8 weeks. Data are shown as mean tumor volume ± SEM. Asterisks indicate a significant difference in ANOVA one-way test (**D**) or two-tailed *t*-test (**A**, **B** and **E**) at the endpoint (*$0.01 < p < 0.05$, **$0.001 < p < 0.01$, ***$0.0001 < p < 0.001$ and ****$p < 0.0001$).

**Combined fulvestrant, CDK4/6i and AKTi is needed to maintain tumor growth inhibition in breast cancer cell lines and tumor xenografts resistant to combined CDK4/6i and fulvestrant.** Previous studies have shown that early adaptation to CDK4/6i can be prevented by combination with endocrine therapy, CDK4/6i and a PI3Ki[22]. However, when tumors progress on combined CDK4/6i and endocrine treatment, the question remains as to whether there is a continued benefit of maintaining CDK4/6i and endocrine therapy and adding an inhibitor of PI3K/AKT-mTOR pathway, or whether CDK4/6i should be switched to a PI3K/AKT-mTOR inhibitor. Here, we assessed the efficacy of fulvestrant, CDK4/6i and AKTi as single agents and in different

combinations in MCF-7 and T47D breast cancer cell lines exhibiting acquired resistance to combined CDK4/6i and fulvestrant treatment (MPF-R and TPF-R, respectively). These cells were generated by continuous exposure to a high dose of fulvestrant and CDK4/6i over 3–4 months, as detailed in "Methods" section. The concentrations of AKTi used in MPF-R and TPF-R models were similar to those used in MCF-7 and T47D fulvestrant-resistant models and were determined based on the highest IC$_{50}$ between parental and resistant cell lines of each cell model (Supplementary Fig. S1C). We found that growth and viability of the MPF-R and TPF-R cell lines were significantly impaired by the triple combination of fulvestrant, CDK4/6i and AKTi (Fig. 4B,

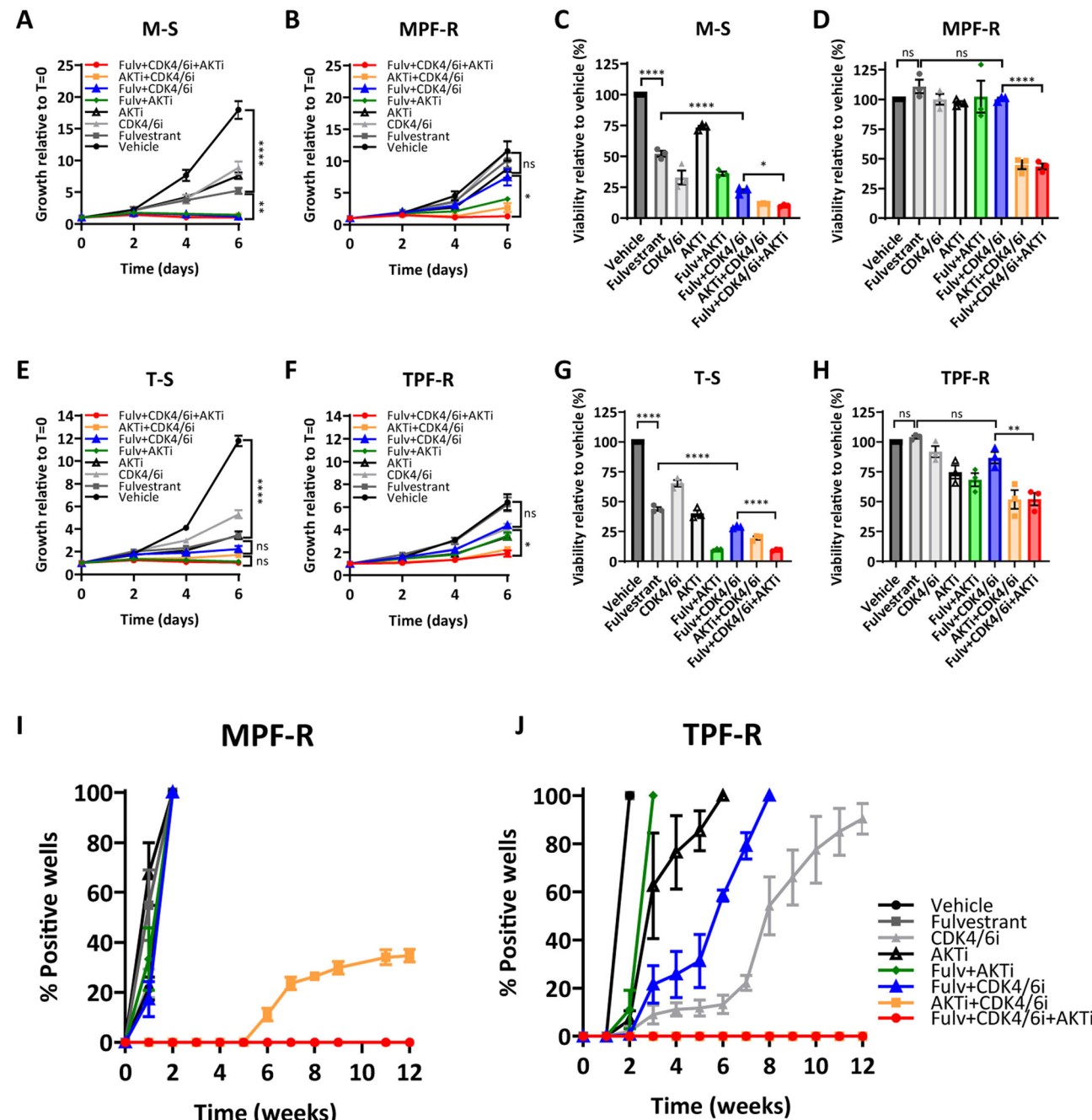

**Fig. 4 Combined fulvestrant, CDK4/6i and AKTi is effective in breast cancer cell lines resistant to combined CDK4/6i and fulvestrant.** The effect of fulvestrant (Fulv, 100 nM), CDK4/6 inhibitor (CDK4/6i, palbociclib, 200 nM) and AKT inhibitor (AKTi, capivasertib, 250–500 nM in the MCF-7 cell model; 100 nM in the T47D cell model), as single agents or in the double and triple combination, was assessed in all cell lines by crystal violet growth assay (**A**, **B**, **E**, and **F**) and CellTiter-Blue viability assay (**C**, **D**, **G**, and **H**) performed over 6 days. Outgrowth of resistant colonies was investigated in MPF-R (**I**) and TPF-R (**J**) cells by weekly evaluation of the percentage of 48 wells at 50% or greater confluence (positive wells) over 12 weeks. Experiments were conducted in three biological replicates and data are shown as mean ± SEM. Asterisks indicate significant differences in one-way ANOVA tests at day 6 (*$0.01 < p < 0.05$, **$0.001 < p < 0.01$, ***$0.0001 < p < 0.001$, and ****$p < 0.0001$).

D, F, and H). Although combined fulvestrant and AKTi was more effective than the standard combination of fulvestrant and CDK4/6i, it was not sufficient to maintain cell growth inhibition in the resistant cell lines MPF-R and TPF-R (Fig. 4B and F). In contrast, fulvestrant combined with either CDK4/6i or AKTi was highly effective in sensitive cells (Fig. 4A, C, E, and G). Moreover, we observed that the triple combination inhibited the growth of resistant colonies in MPF-R and TPF-R cells over the entire 12 weeks of treatment (Fig. 4I–J). Together, these data suggest

that breast cancer cells resistant to the combination of CDK4/6i and endocrine therapy will rapidly progress on fulvestrant and AKTi, but will benefit from the addition of AKTi to the standard combination of CDK4/6i and fulvestrant. Importantly, we evaluated the efficacy of AKTi when combined with other CDK4/6i's, including ribociclib and abemaciclib, in palbociclib-resistant MPF-R and TPF-R cells (Supplementary Fig. S5) and observed that MPF-R and TPF-R cells exhibited a significantly higher IC50 for the three CDK4/6i compared to M-S and T-S cells,

respectively (Supplementary Fig. S5A). The addition of AKTi to the combination of fulvestrant and ribociclib or abemaciclib induced a greater growth inhibition compared to combined CDK4/6i and fulvestrant (Supplementary Fig. S5B-C, respectively). These data suggest that co-targeting CDK4/6 and AKT is efficacious in ER+ breast cancer resistant to the three CDK4/6i's currently clinically approved, which show some differences in the spectrum of target kinases[29]. Currently, there are several ongoing clinical trials evaluating triple combination therapy consisting of endocrine therapy, CDK4/6i, and other inhibitors of the PI3K/ AKT/mTOR pathway in ER+ advanced breast cancer following progression on a CDK4/6i regimen (NCT02871791, NCT02732119). Therefore, we compared the efficacy of the dual PI3K/mTOR inhibitor (PI3K/mTORi) gedatolisib combined with fulvestrant and CDK4/6i with that of the triple combination with AKTi in cell lines resistant to combined CDK4/6i and fulvestrant (Supplementary Fig. S6). We found that the growth inhibition induced in MPF-R and TPF-R cells by the triple combination with dual PI3K/mTORi was similar to that of the triple combination with AKTi (Fig. 4A, B, E, and F), suggesting that these drug combinations have comparable efficacies.

Furthermore, we observed a marked increase in apoptosis and cleaved-PARP levels in MPF-R cells treated with the triple combination compared to the standard fulvestrant and CDK4/6i

combination, although these changes were not observed in TPF-R cells (Fig. 5A–C). We also found that Rb and p-Rb S780 levels were markedly reduced in M-S and T-S cells after treatment with CDK4/ 6i alone and combined with fulvestrant (Supplementary Fig. S7). In addition, we observed that treatment with CDK4/6i alone and combined with fulvestrant decreased Rb levels to a lower extent in MPF-R and TPF-R cells than in the parental cells (Supplementary Fig. S7A). p-Rb S780 baseline levels were also significantly lower in MPF-R and TPF-R cells compared to the parental cells (Supplementary Fig. S7B). These data support reduced dependence on the cyclin D1-CDK4/6 pathway in MPF-R and TPF-R cells, as previously shown in other cell line models resistant to CDK4/6i[22]. We observed a marked reduction of p-PRAS40 and p-S6 expression when MPF-R and TPF-R cells were treated with AKTi (Fig. 5D), consistent with PI3K/AKT-mTOR pathway blockade. Furthermore, MPF-R exhibited higher levels of p-AKT and similar levels of total AKT compared to parental M-S cells, while p-AKT expression was slightly lower and total AKT was higher in TPF-R compared to parental T47D sensitive T-S cells (Fig. 5E).

To further investigate our in vitro findings (Fig. 4A–J), we compared the efficacy of the triple and double combinations in MPF-R tumor xenografts. MPF-R cells were orthotopically implanted in the mammary fat pad of mice and, when tumors reached approximately 100 mm³, treatment with fulvestrant

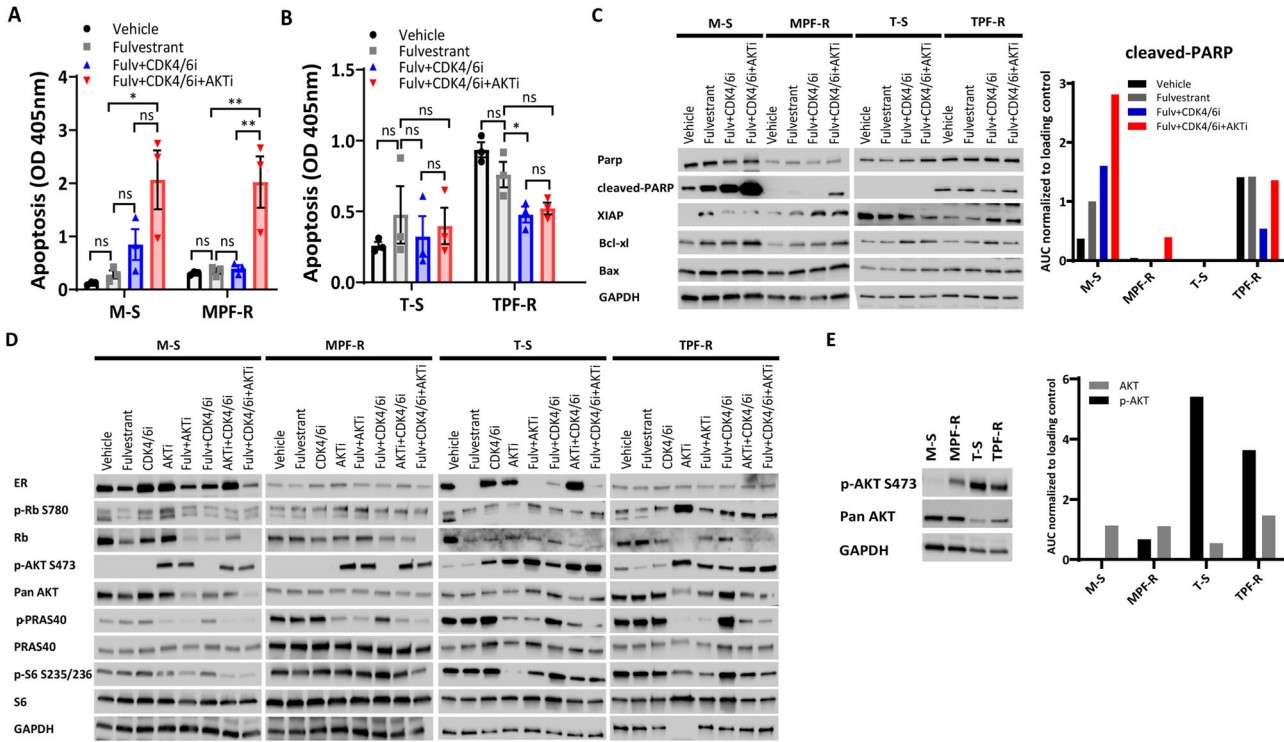

**Fig. 5 Combined targeting of ER, CDK4/6 and AKT efficiently inhibits cyclin D/CDK4-6/Rb and PI3K/AKT-mTOR pathways in breast cancer cells resistant to combined CDK4/6i and fulvestrant.** Apoptotic levels were determined by evaluating the presence of cytoplasmic nucleosomes in MCF-7 (**A**) and T47D (**B**) cell models using an ELISA cell death detection assay. Data are shown with error bars representing mean ± SEM of three biological replicates. Asterisks indicate significant differences in one-way ANOVA tests (*0.01 < $p$ < 0.05, **0.001 < $p$ < 0.01, ***0.0001 < $p$ < 0.001, and ****$p$ < 0.0001). **C** Western blot analysis of apoptosis markers in both models resistant to combined fulvestrant and CDK4/6i. Densitometry analysis of Western blot bands of cleaved-PARP was performed using ImageJ software. Data are shown as the area under the curve (AUC) normalized to loading control. **D** Western blot analysis of key signal transduction proteins in breast cancer cell models resistant to combined CDK4/6i and fulvestrant therapy. **E** Western blotting analysis of p-AKT (S473) and total AKT expression in both breast cancer models resistant to combined fulvestrant and CDK4/6i. Densitometry analysis of Western blot bands of p-AKT (S473) and total AKT was performed using ImageJ software. Data are shown as the area under the curve (AUC) normalized to loading control. GAPDH was used as a loading control for Western blotting analysis. For all Western blots, a representative of two biological replicates is shown. Both apoptosis assay and harvesting protein for Western blotting analysis were performed 3 days after treatment with fulvestrant (Fulv, 100 nM), CDK4/6 inhibitor (CDK4/6i, palbociclib, 200 nM), and AKT inhibitor (AKTi, capivasertib, 250 nM in the MCF-7 cell model; 100 nM in the T47D cell model).

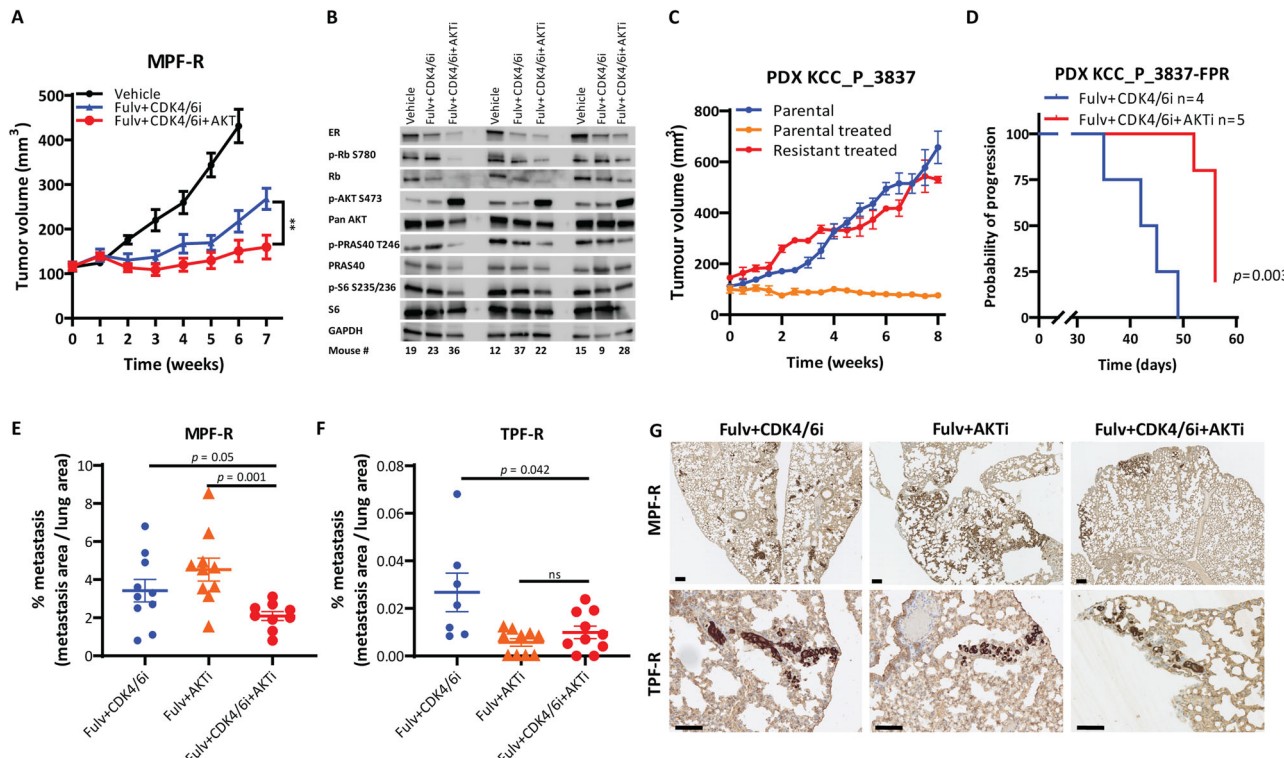

**Fig. 6 Combined fulvestrant, CDK4/6i and AKTi is effective in tumor xenografts resistant to combined CDK4/6i and fulvestrant. A** Tumor growth curves of orthotopic MPF-R tumors treated with a CDK4/6 inhibitor (CDK4/6i, palbociclib, 50 mg/Kg bodyweight) combined with fulvestrant (Fulv, 100 mg/Kg bodyweight; $N = 9$) or in combination with both AKT inhibitor (AKTi, capivasertib, 100 mg/Kg body weight) and fulvestrant (Fulv, 100 mg/Kg bodyweight; $N = 10$), or vehicle (castor oil and 25% w/v HPB cyclodextrin; $N = 10$). CDK4/6i and AKTi were administered by oral gavage once daily for 5 days a week, whereas fulvestrant was administered subcutaneously once a week. Treatment was initiated when tumors reached 100 mm$^3$ and continued for up to 7 weeks. Mice from the control group were sacrificed and tumors excised on week 6 due to their large size, while mice from double and triple combination groups were sacrificed and tumors excised on week 7. Data are shown as mean tumor volume ± SEM. Asterisks indicate significant differences in the two-tailed $t$-test at the endpoint (*$0.01 < p < 0.05$, **$0.001 < p < 0.01$, ***$0.0001 < p < 0.001$, and ****$p < 0.0001$). **B** Western blot analysis of key signal transduction proteins in 3 tumors of each treatment group excised when mice were sacrificed. GAPDH was used as a loading control. A representative of two independent experiments is shown. **C** Tumor volumes over time of the parental PDX KCC_P_3837 untreated (blue) and treated with combined CDK4/6i palbociclib (25 mg/kg in 2.5% DMSO, 25% β-cyclodextrin, 5 days per week by oral gavage) and fulvestrant (100 mg/kg in castor oil, once weekly via subcutaneous injection) (orange), and of the derivative PDX KCC_P_3837-FPR resistant to combined palbociclib and fulvestrant (red) at the third passage of continuous exposure to combined palbociclib and fulvestrant. **D** Kaplan-Meier survival plot of progression of PDX KCC_P_3837-FPR (resistant to combined CDK4/6i and fulvestrant) under treatment with combined CDK4/6i and fulvestrant with or without AKTi capivasertib (100 mg/kg in 2.5% DMSO, 25% β-cyclodextrin, 5 days per week by oral gavage) ($N = 5$ and $N = 4$, respectively). Progression was defined as tumors growing to at least 5 mm in the shortest dimension. A two-sided $p$-value ($p < 0.05$) was calculated using log-rank testing. **E**, **F** Evaluation of metastasis area > 2500 μm$^2$ relative to lung area at the endpoint (6 weeks) using an experimental metastasis model. MPF-R tumors were treated with fulvestrant (100 mg/Kg body weight) combined with CDK4/6i (palbociclib, 25 mg/Kg bodyweight; $N = 10$), fulvestrant combined with AKTi (capivasertib, 100 mg/Kg bodyweight; $N = 10$), or triple combination ($N = 9$). TPF-R tumors were treated with the same dosage of fulvestrant and CDK4/6i and 50 mg/Kg bodyweight of AKTi ($N = 7$ in fulvestrant and CDK4/6i group, $N = 10$ in fulvestrant and AKTi and $N = 10$ in triple combination group). CDK4/6i and AKTi were administered by oral gavage once daily for 5 days a week, while fulvestrant was administered subcutaneously once a week. Treatment was initiated 3 days before injection of cells in the tail vein and continued for up to 6 weeks. Data are shown as mean ± SEM. Significant differences were evaluated by the two-tailed Mann-Whitney test. **G** Representative micrographs of MPF-R and TPF-R tumors in mice lungs of each treatment group showing cytokeratin expression by immunohistochemistry (scale bars, 200 μm).

(100 mg/Kg) and CDK4/6i (50 mg/Kg) with or without AKTi (100 mg/Kg) was initiated and continued for 7 weeks. Although the double combination significantly inhibited tumor growth of MPF-R tumors compared to vehicle, it failed to induce tumor regression (Fig. 6A), in contrast with the profound tumor regression observed in 182R-1 fulvestrant-resistant tumors (Fig. 3E). This supports the reduced sensitivity of MPF-R cells to the standard combination of fulvestrant and CDK4/6i. More importantly, the triple combination almost completely inhibited tumor growth over the 7 weeks of treatment, while tumors treated with the double combination started to regrow after 5 weeks (Fig. 6A). Furthermore, a statistically significant difference in tumor size and weight were observed between mice treated with the triple vs. double combinations at the

endpoint, as evaluated by the parametric $t$-test and the non-parametric Wilcoxon test (Fig. 6A and Supplementary Fig. S8A). We also found a statistically significant difference in tumor growth rate between double and triple combinations using linear mixed-effects models (GR 13.36, $p < 0.0001$, CI 8.41–18.29).

In addition, we observed a significant change of ER, p-Rb, p-AKT, p-PRAS40, and p-S6 levels in MPF-R tumors treated with the triple combination (Fig. 6B and Supplementary Fig. S8B), similar to our in vitro findings (Fig. 5D). Although no significant change in cleaved caspase-3 expression was observed, decreased expression of the proliferation marker Ki67 was found in MPF-R tumors treated with the triple combination compared to double combination (Supplementary Fig. S8C–D), which is also in line with our in vitro

findings (Fig. 4A–D). Furthermore, HE staining showed that tumors treated with combined CDK4/6i and fulvestrant primarily consisted of vital tumor tissue, while tumors treated with the triple combination were smaller and contained larger areas of degeneration and reactive fibrosis (within the indicated circles) or smaller areas of degeneration and lipid infiltration surrounded by vital tumor tissue (Supplementary Fig. S8C).

Next, we evaluated the efficacy of the triple combination with AKTi in a PDX model resistant to combined CDK4/6i and fulvestrant (KCC_P_3837-FPR), generated through continuous exposure of PDX KCC_P_3837 tumors to combined palbociclib and fulvestrant over three passages in mice (Fig. 6C). We observed that PDX KCC_P_3837-FPR mice treated with the triple combination showed a statistically significant ($p = 0.003$) increase in progression-free survival (PFS), with progression defined as tumor width $\geq$ 5 mm, compared to mice in the combined CDK4/6i and fulvestrant treatment group (Fig. 6D). Decreased Ki67 expression was observed in tumors treated with the triple combination compared to the combined CDK4/6i and fulvestrant (Supplementary Fig. S8D). Furthermore, HE staining showed that PDX tumors treated with the triple combination contained large areas of degeneration and reactive fibrosis (within the indicated circles) surrounded by vital tumor tissue. In contrast, only small areas, if any, of degeneration and reactive fibrosis were observed in tumors treated with combined CDK4/6i and fulvestrant, and the vital tumor tissue area was much larger in these tumors (Supplementary Fig. S8D). Finally, we performed two sets of animal experiments to analyze the effect of the triple combination, combined fulvestrant and CDK4/6i or combined fulvestrant and AKTi in MPF-R and TPF-R cells using an experimental metastasis model. To mimic the scenario wherein tumor cells under the selective pressure of the treatment enter the circulation and subsequently develop metastasis in lungs and liver, we pre-treated the cells in vitro for 3 days, which inhibited the signaling pathways without affecting viability, before injection of cells into the tail vein. The triple combination-treated group contained significantly fewer metastasis (defined, approximately, as tumor area > 2500 $\mu m^2$) compared to the combined fulvestrant and CDK4/6i-treated group at 6 weeks of treatment in both MPF-R ($p = 0.050$, Fig. 6E) and TPF-R ($p = 0.042$, Fig. 6F) models. For the TPF-R model, the number and size of metastasis were small, likely due to the slow growth rate of these cells (Fig. 6G). Overall, the metastasis was smaller in the triple combination-treated group compared to the fulvestrant and CDK4/6i combination-treated group in both MPF-R ad TPF-R models (Fig. 6G). Furthermore, the triple combination-treated group contained significantly fewer metastasis than the fulvestrant and AKTi combination-treated group in the MPF-R model ($p = 0.001$) (Fig. 6E and G). Together, our findings show that the triple combination of fulvestrant, CDK4/6i and AKTi effectively inhibits the growth of tumors that expand on combined fulvestrant and CDK4/6i treatment.

**High expression of p-AKT correlates with shorter PFS in ER+ advanced breast cancer treated with combined fulvestrant and CDK4/6i.** Next, we investigated the clinical relevance of higher levels of p-AKT in MCF-7 and ZR-75-1 breast cancer cells resistant to fulvestrant (Fig. 2F) and combined fulvestrant and CDK4/6i (MPF-R cells) (Fig. 5E). p-AKT levels in full sections of metastatic lesions of ER+breast cancer patients treated with combined endocrine therapy and CDK4/6i in the advanced setting were evaluated by immunohistochemistry. Initially, we determined the cut-off value of p-AKT scoring by evaluating p-AKT levels in a pilot cohort ($N = 17$), which included patients treated at Odense University Hospital, Denmark, with a

metastatic biopsy obtained in 2019. Although patients included in the pilot cohort had a short clinical follow-up, we determined a cut-off value that showed survival significance in Kaplan-Meier curves (cut-off $\geq$ 150; $p = 0.03$; Supplementary Fig. S9) and, thus, we selected this value as the cut-off to stratify patients into high and low p-AKT in the validation cohort ($N = 84$, metastatic biopsy obtained in 2017–2018). Clinical and pathological characteristics of the primary tumor and metastatic disease of patients from both cohorts are shown in Supplementary Table S1 and Table 1, respectively. Although $\chi2$ and Fisher's exact tests identified a statistically significant difference in primary tumor size between low and high p-AKT groups, this difference was found in both pilot and validation cohorts (Supplementary Table S1). No other differences in clinical and pathological characteristics of primary tumors or metastatic disease were observed in the pilot and validation cohorts (Supplementary Table S1 and Table 1). Kaplan-Meier curves of the validation cohort showed significantly ($p = 0.04$) lower PFS in the high p-AKT group (H-score $\geq$ 150; 9.87 months) compared with the low p-AKT group (H-score < 150; 15.37 months), corresponding to a 6-month increase in the median time to progression (Fig. 7A). Univariate Cox's proportional hazards regression analysis showed that only p-AKT status (HR 2.07, 95% CI of the ratio, 1.00-4.29, $p = 0.049$; Supplementary Table S2) and line of therapy (HR 3.05, 95% CI of the ratio, 1.61–5.79, $p = 0.001$; Supplementary Table S2) were prognostic factors for PFS for patients treated with combined CDK4/6i and endocrine therapy. Evaluation of the distribution of the metastatic variables included in the Cox regression analysis between p-AKT-low vs. p-AKT-high groups showed that age, number of metastasis, time to recurrence, line of therapy, and site of relapse are evenly distributed between the two groups (Supplementary Table S3). Representative immunohistochemistry stainings of low (Fig. 7B-C) and high (Fig. 7D–E) p-AKT levels are shown. Interestingly, we found that not only patients with *PIK3CA*-mutated tumors exhibited high p-AKT levels, and patients with *PIK3CA* wild-type tumors exhibited low p-AKT levels, but the opposite was also observed, suggesting that the level of p-AKT is independent of *PIK3CA* mutation status.

## Discussion

CDK4/6i has demonstrated impressive efficacy in combination with an aromatase inhibitor or fulvestrant in ER+ advanced breast cancer. However, not all patients benefit significantly from these combination treatments, and even those who do their tumors are expected to eventually progress. Therefore, there is a need to evaluate better and more rational targeted combinations to prevent or overcome resistance to standard CDK4/6i and endocrine therapy combination and identify biomarkers for the selection of patients who will benefit from these targeted combinations. In this study, we show that standard combined CDK4/6i and endocrine therapy does not efficiently suppress the growth of ER+ breast cancer cell lines and tumor xenografts resistant to fulvestrant, while the addition of AKTi results in profound growth inhibition. Furthermore, the triple combination of CDK4/6i, AKTi, and endocrine therapy efficiently suppressed the growth and reduced metastasis of tumors resistant to standard CDK4/6i and endocrine therapy combinations. This was demonstrated in several orthotopic and experimental metastasis models, as well as in a PDX model resistant to combined CDK4/6i and fulvestrant.

It is well known that growth factor-mediated activation of AKT can regulate ER signaling, resulting in ligand-independent activation of ER genomic pathway[30,31]. Furthermore, high AKT levels have been shown to modulate ER binding and estrogen-regulated gene expression[32]. Activation of AKT has also been associated with resistance to endocrine therapy[33]. Together, these

**Table 1 Clinical and pathological characteristics of ER+ breast cancer patients with advanced disease treated with combined CDK4/6i and endocrine treatment from pilot and validation cohorts according to p-AKT levels.**

| Parameters | Pilot | | | | Validation | | | | Pilot vs. validation | | | |
|---|---|---|---|---|---|---|---|---|---|---|---|---|
| | p-AKT low | p-AKT high | N | $p^a$ | p-AKT low | p-AKT high | N | $p^a$ | pilot | validation | N | $p^a$ |
| Age at starting CDK4/6i | | | | | | | | | | | | |
| ≤50 | 1 | 1 | 2 | 0.33 | 5 | 2 | 7 | 0.63 | 2 | 7 | 9 | 0.65 |
| >50 | 13 | 2 | 15 | | 62 | 15 | 77 | | 15 | 77 | 92 | |
| Site of relapse[b] | | | | | | | | | | | | |
| Soft tissue | 9 | 1 | 10 | 0.53 | 20 | 5 | 25 | 0.13 | 10 | 25 | 35 | 0.37 |
| Bone | 3 | 2 | 5 | | 34 | 5 | 39 | | 5 | 39 | 44 | |
| Viscera | 2 | 0 | 2 | | 13 | 7 | 20 | | 2 | 20 | 22 | |
| No metastatic sites | | | | | | | | | | | | |
| 1 | 4 | 1 | 5 | 0.86 | 18 | 6 | 24 | 0.62 | 5 | 24 | 29 | 0.16 |
| 2 | 7 | 1 | 8 | | 19 | 3 | 22 | | 8 | 22 | 30 | |
| ≥3 | 3 | 1 | 4 | | 30 | 8 | 38 | | 4 | 38 | 42 | |
| Chemotherapy[c] | | | | | | | | | | | | |
| No | 9 | 1 | 10 | 0.54 | 37 | 8 | 45 | 0.60 | 10 | 45 | 55 | 0.79 |
| Yes | 5 | 2 | 7 | | 30 | 9 | 39 | | 7 | 39 | 46 | |
| Time to recurrence (years) | | | | | | | | | | | | |
| ≤5 | 7 | 3 | 10 | 0.28 | 28 | 5 | 33 | 0.35 | 10 | 33 | 43 | 0.29 |
| 1–10 | 4 | 0 | 4 | | 16 | 7 | 23 | | 4 | 23 | 27 | |
| >10 | 3 | 0 | 3 | | 23 | 5 | 28 | | 3 | 28 | 31 | |
| Total | 14 | 3 | 17 | | 67 | 17 | 84 | | 17 | 84 | 101 | |

[a]Two-sided $\chi$2 or Fisher's exact test.
[b]Site of relapse of the metastatic lesion used to evaluate p-AKT expression.
[c]Includes chemotherapy administrated in the adjuvant and metastatic settings.

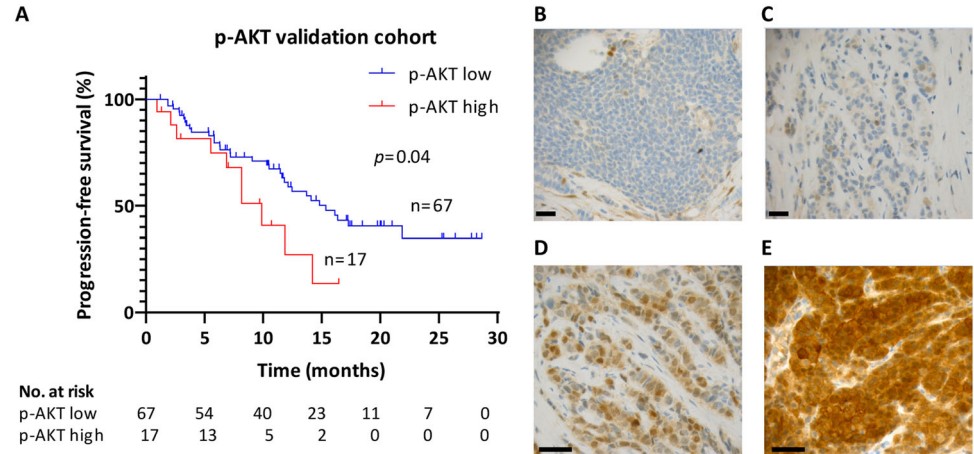

**Fig. 7 p-AKT expression correlates with PFS in ER+ metastatic breast cancer patients treated with combined CDK4/6i and endocrine therapy.**
**A** Kaplan-Meier plots evaluating progression-free survival (PFS) according to p-AKT (S473) levels in ER+ metastatic lesions from a validation cohort of ER+breast cancer patients treated with combined CDK4/6i and endocrine therapy in the advanced setting. A two-sided *p*-value (*p* < 0.05) was calculated using log-rank testing. Representative micrographs of all breast cancer metastasis sections showing low p-AKT expression (H-score < 150, **B** and **C**) or high p-AKT expression (H-score ≥ 150, **D** and **E**; scale bars, 100 μm).

data have led to the clinical development of AKTi in combination with endocrine therapy for ER+ breast cancer. Indeed, the clinical trial FAKTION has recently investigated the addition of AKTi capivasertib to fulvestrant for postmenopausal women with ER+ breast cancer who progressed on an AI and showed that combined capivasertib and fulvestrant significantly extended PFS compared to fulvestrant monotherapy[19]. However, in our study, we observed that the effect of treatment with fulvestrant combined with AKTi is limited, and resistance quickly develops for MCF-7 and T47D cells resistant to fulvestrant monotherapy. The addition of CDK4/6i was required to suppress the growth of resistant clones. Nevertheless, endocrine treatment-sensitive cells showed durable tumor growth inhibition with both fulvestrant plus AKTi or fulvestrant plus CDK4/6i. Although we have only

examined the efficacy of a triple combination using fulvestrant as the endocrine agent, it has been previously shown that high levels of AKT activity confer resistance to letrozole and anastrozole[34,35], and thus we believe that triple combination with AKTi, CDK4/6i, and AIs might be effective in all AI-resistant tumors.

Although the alpha-specific PI3Ki alpelisib has recently been shown to significantly improve PFS in *PIK3CA*-mutated ER+ advanced breast cancer that progressed on previous endocrine therapy, this PI3Ki is not active in *PIK3CA* wild-type and *PTEN* null tumors[17]. In contrast, capivasertib has demonstrated efficacy in tumors regardless of *PIK3CA* and *PTEN* status[36]. Indeed, the FAKTION clinical trial showed that the *PIK3CA* mutation did not affect the response to combined capivasertib and fulvestrant in ER+ metastatic breast cancer[19]. These findings concur with the

preclinical observations in our study that both *PIK3CA* mutant and *PTEN* wild-type (MCF-7 and T47D) and *PIK3CA* wild-type and *PTEN* null (ZR-75-1) cell lines benefited from the addition of AKTi to the standard combination of fulvestrant and CDK4/6i. Importantly, a triple combination of fulvestrant, CDK4/6i and AKTi was required for long-term growth inhibition of our fulvestrant-resistant cells derived from MCF-7 and T47D cell line models (*PIK3CA* mutant). Nevertheless, PI3K controls additional pathways that are independent of AKT, such as the ERK signaling pathway, and therefore AKT blockage might not be able to inhibit tumor growth as efficiently as a specific PI3Ki in all *PIK3CA* mutant tumors[37]. Although significant toxicity was reported in the FAKTION clinical study with combined capivasertib and fulvestrant, particularly diarrhea, rash, and hyperglycemia, these side effects are also observed with other drugs targeting regulators of the PI3K/AKT-mTOR pathway and do not overlap with the palbociclib hematological toxicity profile, indicating that the side effects associated with the addition of CDK4/6i to this double combination might be clinically manageable[19].

Previous studies have suggested that simultaneous blockade of PI3K and CDK4/6 is needed to completely inhibit cyclin D1[22]. Indeed, upon treatment with CDK4/6i, upregulation of AKT with subsequent accumulation of cyclin D1 and sustained expression of cyclin E2 and CDK2 has been observed, which promotes progression into S-phase contributing to resistance. Initial triple combination of endocrine therapy, CDK4/6i and PI3Ki in vitro and in patient-derived xenografts achieved greater cell cycle arrest, decreased cyclin E2 and CDK2 expression with subsequent induction of apoptosis, and induced greater tumor regression than each inhibitor alone[22]. However, this effect was not reproduced in cell lines with acquired resistance to CDK4/6i, suggesting that the combination of CDK4/6i and PI3Ki may more effectively delay resistance in CDK4/6i-naïve tumors[22]. However, preliminary results from the phase II clinical trial (BYLieve) with alpha-selective PI3Ki alpelisib and fulvestrant suggest that this combination may also be efficacious in patients with *PIK3CA* mutations that were previously treated with CDK4/6i, although it is too early for firm conclusions[38]. In line with these findings, we show herein that triple combination with fulvestrant, CDK4/6i and another PI3K/AKT-mTOR inhibitor, AKTi, efficiently suppresses the growth of cell lines and reduces tumor progression in cell-derived and patient-derived xenografts with acquired resistance to CDK4/6i and fulvestrant combination. In addition, we found that triple combination with the dual PI3K/mTORi gedatolisib, the recent evaluation of which in a phase 1b clinical trial in combination with CDK4/6i and endocrine therapy demonstrated tolerability and preliminary efficacy in ER+ advanced breast cancer patients[39], showed comparable efficacy to the triple combination with AKTi in inhibiting the growth of cell lines resistant to combined CDK4/6i and fulvestrant. Interestingly, it has been shown that upregulation of AKT and non-AKT targets of PDK1 lead to aberrant cell-cycle progression in ribociclib-resistant cell lines[26]. In addition, it has been recently found that *PTEN*-deficient cells exhibit cross-resistance to CDK4/6i and alpha-selective PI3Ki, mediated by AKT activation, which can be overcome by treatment with AKTi[40]. It is noteworthy that in this study, some CDK4/6i-resistant tumors retained *PTEN*, indicating that other mechanisms also mediate resistance to CDK4/6i[40]. Nevertheless, the data presented in our study showed that *PTEN* wild-type cell lines and tumor xenografts resistant to CDK4/6i and fulvestrant (MPF-R and TPF-R cells) benefited from AKT inhibition, demonstrating the need for clinical trials of AKTi in combination with standard CDK4/6i and endocrine therapy in the post CDK4/6i-setting independent of the tumor *PTEN* status. A phase Ib/III trial has recently begun to evaluate the AKTi capivasertib plus palbociclib and fulvestrant vs. palbociclib and

fulvestrant in ER+ locally advanced, unresectable, or metastatic breast cancer (CAPItello-292, NCT04862663), further supporting the clinical relevance of our study.

Furthermore, we observed in this study that MCF-7 and ZR-75-1 breast cancer cell lines resistant to endocrine therapy exhibited higher levels of p-AKT compared to sensitive cells, in line with previous findings of a significant increase in p-AKT and high AKT kinase activity in antiestrogen-resistant cell lines[41]. In addition, it has previously been shown that breast cancer patients with p-AKT-positive tumors correlated with worse clinical outcomes on endocrine therapy compared to patients with p-AKT-negative tumors[42]. Importantly, our MCF-7-derived breast cancer cell line resistant to the combination of fulvestrant and CDK4/6i also expressed higher p-AKT levels compared to the respective parental cell line. Indeed, activation of the PI3K/AKT-mTOR pathway by PDK1-mediated phosphorylation of AKT (S477/T479) in ribociclib-resistant breast cancer cells has been previously shown[26]. However, the role of p-AKT as a prognostic or predictive biomarker in CDK4/6i-treated patients has not been investigated. Although many biomarkers of resistance to CDK4/6i have been evaluated in preclinical and clinical studies, including Rb loss or mutation, p16 loss, *PIK3CA* mutation, *FAT1* mutation, aberrant FGFR pathway, CDK6, cyclin D1, and cyclin E amplification, and other D-cyclin-activating features, biomarkers with clinical validity have yet to be identified and represent an unmet need[43–49]. Although a comprehensive analysis of the clinical significance of p-AKT is not possible in our study due to the small sample size, we believe that our data clearly indicate that the level of p-AKT is associated with CDK4/6 resistance and provide a firm basis and an interesting perspective for future studies. Furthermore, it would have been interesting to determine the AKT mutation status of the clinical samples and evaluate the correlation with clinical outcome, however, genotyping of metastatic breast cancers has just recently become available at our hospital and only on selected patients. Nevertheless, our data suggest that patients with metastasis exhibiting high levels of p-AKT are associated with a worse prognosis on treatment with standard CDK4/6i and endocrine therapy and may benefit from the addition of an AKTi to improve survival.

## Methods

**Cell lines and anti-tumor agents**. The original MCF-7 and T47D cell lines were obtained from the Breast Cancer Task Force Cell Culture Bank, Mason Research Institute, and the original ZR-75-1 cell line was obtained from the American Type Culture Collection (ATCC). Fulvestrant-resistant cell line 182R-1 was established from MCF-7/S0.5 cells (designated as MCF-7 throughout the manuscript) by extended treatment with 100 nM of fulvestrant[50]. MCF-7 cells were routinely propagated in phenol red-free Dulbecco's Modified Eagle Medium DMEM/F12 (Gibco) supplemented with 1% glutamine (Gibco), 1% heat-inactivated fetal bovine serum (FBS; Sigma-Aldrich), and 6 ng/ml insulin (Sigma-Aldrich). 182R-1 cells were maintained in the same growth medium as MCF-7 cells supplemented with 100 nM fulvestrant. MCF-7-derived cell lines resistant to combined CDK4/6i and fulvestrant (MPF-R) were developed from 182R-1 cells by prolonged treatment (4 months) with 150–200 nM of CDK4/6i and 100 nM of fulvestrant and maintained in the same growth medium as 182R-1 cells supplemented with 200 nM CDK4/6i. MCF-7-sensitive cells grown in parallel with MPF-R cells were designated M-S. T47D cells were maintained in RPMI 1640 media without phenol red supplemented with 1% glutamine, 5% FBS, and 8 μg/ml insulin. T47D-derived fulvestrant-resistant cell lines (T47D R) were established from the T47D cell line by long-term treatment with 100 nM fulvestrant[51]. T47D cells resistant to fulvestrant and CDK4/6i (TPF-R) were established from T47D R cells by long-term treatment (3 months) with 100 nM fulvestrant and 150–200 nM CDK4/6i and maintained in the same growth medium as T47D R cells supplemented with 200 nM CDK4/6i. T47D-sensitive cells grown in parallel with TPF-R were designated T-S. ZR-75-1 cells were routinely propagated in Roswell Park Memorial Institute (RPMI) 1640 medium (Gibco) supplemented with 10% FBS, 1% HEPES (Gibco), and 1% Penicillin/Streptomycin (Gibco). ZR-75-1 cells were used to establish the fulvestrant-resistant cell line ZR-75-1 R by long-term (8 weeks) exposure to increasing concentrations of fulvestrant from 100 pM to a final concentration of 100 nM. ZR-75-1 R cells were grown in the same growth media as the parental cell line supplemented with 100 nM fulvestrant. Cells were grown in a humidified

atmosphere of 5% $CO_2$ at 37 °C. All cell lines underwent DNA authentication using Cell ID™ System (Promega) and mycoplasma testing (Lonza) before the described experiments. Fulvestrant (ICI 182,780, Tocris) was dissolved in ethanol 96%, AKTi capivasertib (HY-15431, MedchemExpress), CDK4/6i ribociclib succinate hydrate (HY-15777C, MedchemExpress), CDK4/6i abemaciclib (HY-16297A, Medchem-Express), and dual PI3K/mTORi gedatolisib (#HY-10681, MedchemExpress) were dissolved in DMSO (Sigma-Aldrich) and CDK4/6i palbociclib isothiocyanate (HY-A0065, MedchemExpress) was dissolved in water. The concentrations of CDK4/6i, AKTi, and dual PI3K/mTORi to be used for in vitro experiments were determined based on the $IC_{50}$ for each cell line model.

### Western blotting.
Whole-cell extracts were obtained using RIPA buffer (50 mM Tris HCl (pH 8), 150 mM NaCl (pH 8), 1% IgePAL 630, 0.5% sodium deoxycholate, 0.1% SDS) containing protease and phosphatase inhibitors Complete and PhosSTOP (Roche). The protein concentration of the lysate samples was determined using Pierce BCA Protein Assay Kit (Thermo Fisher Scientific) and the optical density (OD) was measured at 562 nm in the microplate reader Paradigm (Beckman Coulter). 5–30 μg of total protein lysate was loaded on a 4–20% SDS-PAGE gel (Bio-Rad) under reducing conditions and electroblotted onto a PVDF transfer membrane (Bio-Rad). Membranes were blocked in Tris-buffered saline (TBS), 0.1% Tween-20 (Sigma-Aldrich) containing 5% non-fat dry milk powder (Sigma-Aldrich) for one hour at room temperature. The following primary antibodies were used according to the manufacturer´s protocol: anti-ER (RM-9101-S1, 1:500) from Thermo Fisher Scientific; anti-p-Rb S780 (3590, 1:1000), anti-Rb (9309, 1:1000), anti-p-AKT S473 (4060, 1:1000), anti-AKT (pan) (4685, 1:1000), anti-p-PRAS40 T246 (2997, 1:1000), anti-PRAS40 (2691, 1:1000), anti-p-S6 S235/236 (2211, 1:1000), anti-S6 (2217, 1:1000), anti-cleaved PARP (9541, 1:250), anti-PARP (9532, 1:1000), anti-Xiap (2042, 1:500), anti-Bcl-xl (2762, 1:1000), anti-Bax (2772, 1:1000) from Cell Signaling; and anti-GAPDH (sc-32233, 1:20000) from Santa Cruz Biotechnology as loading control. Secondary antibodies horseradish peroxidase (HRP)-conjugated goat anti-mouse (#P0447, Dako, 1:5000) and HRP-conjugated goat anti-rabbit (#P0448, Dako, 1:5000) were incubated in a blocking buffer for one hour at room temperature. Membranes were developed with Clarity™ Western ECL Substrate (Bio-Rad) and visualized on Fusion-Fx7-7026 WL/26MX instrument (Vilbaer).

### Cell growth, viability, and apoptosis assays.
Cells were seeded at 750–4000 cells/well in 96-well plates and allowed to attach for 24 h before drugs or vehicles were added. Evaluation of cell growth was performed using crystal violet-based colorimetric assay[52] and the OD was analyzed at 570 nm in Paradigm reader. Cell viability was evaluated by CellTiter-Blue (Promega) according to the manufacturer´s instructions and fluorescence was measured at 560/590 nm in Paradigm reader. Apoptosis was assessed using the Cell Death Detection ELISA$^{Plus}$ kit (Roche) according to the manufacturer's instructions and the OD was analyzed at 405/490 nm in Paradigm reader. For the colony outgrowth assay, cells were seeded (700–1500 cells/well) in 96-well plates (48 wells/treatment). Medium with the vehicle, single drug, or drug combinations was changed once a week and positive wells were scored weekly as >50% confluent[53].

### Drug interaction analysis.
Cells were seeded at 2500 cells/well and allowed to attach for 24 h before drugs or vehicles were added. Cells were treated with increasing doses of palbociclib, capivasertib, and fulvestrant or an equipotent combination of the inhibitors and incubated at 5% $CO_2$ and 37 °C for 3 days. Cell growth was evaluated using crystal violet-based colorimetric assay and interactions were calculated with Compusyn software (ComboSyn, Inc.), based on the combination index (CI) equation from Chou-Talalay method[54]. Drug interaction was scored as follows: CI = 1 is additive, CI < 1 is synergistic, CI > 1 is antagonistic.

### Xenograft studies.
For primary tumor growth, MCF-7, 182R-1, and MPF-R cells were harvested using trypsin (Sigma-Aldrich) and $1 \times 10^6$ cells were resuspended in 50 μl of extracellular matrix (ECM) from Engelbreth-Holm-Swarm sarcoma (Sigma-Aldrich) and injected orthotopically into the mammary fat pad of 7-week-old female NOG CIEA mice (Taconic) without exogenous estrogen supplements. When tumors reached a certain size (indicated in the figure legend), mice were weighed and randomized into treatment groups. For the experimental metastasis model, MPF-R and TPF-R were harvested using trypsin, and $1 \times 10^6 – 2 \times 10^6$ cells were resuspended in 100–400 μl of culture media. $1 \times 10^6$ cells were inoculated into the lateral tail vein of 7-week-old female NOG CIEA mice. Fulvestrant (Faslodex, AstraZeneca) was formulated at 20 mg/ml in castor oil (Sigma-Aldrich) and administrated once a week subcutaneously. Capivasertib and palbociclib, both from MedchemExpress, were formulated at 20 and 10 mg/ml, respectively, in 25% w/v HPB cyclodextrin (Sigma-Aldrich) with sonication and administered 5 days a week by oral gavage. Treatment was continued for 5–8 weeks. For the orthotopic model tumor volume was calculated as: tumor volume = $0.5 \times$ (length) $\times$ (width)$^2$. For the experimental metastasis model, visualization and quantification of metastasis in the lungs at the endpoint were performed by immunohistochemistry using anti-cytokeratin antibody on full lung sections from 3 different depths. Slides were scanned and analyzed using ndp.view 2.3.14 software (Hamamatsu). The amount of metastasis (defined, approximately, as >2500 μm$^2$) relative to lung area was

subsequently determined by ImageJ analysis[55] in a blinded setup. A similar approach was used to quantify the metastatic burden for the TPF-R model, but because the metastasis was quite small in this model, they were measured manually in a blinded setup. All animal experiments were approved by the Experimental Animal Committee of The Danish Ministry of Justice and were performed at the animal core facility at the University of Southern Denmark. Mice were housed under pathogen-free conditions with ad libitum food and water. The light/dark cycle was 12 h light/dark, with lights turned on from 6 a.m. to 6 p.m. Housing temperature was 21 + 1 °C and relative humidity 40–60%.

### PDX model.
KCC_P_3837 was derived from an untreated grade 3, ER+, PR+, HER2− primary invasive ductal carcinoma. KCC_P_3837-FPR (resistant to combined CDK4/6i and fulvestrant) was generated through continuous exposure of KCC_P_3837 tumors to the combination of palbociclib (50 mg/kg in water, 5 days per week by oral gavage) and fulvestrant (5 mg/kg in peanut oil, once weekly via subcutaneous injection) over several passages in mice. At each passage, tumors were established to a width of 5 mm before treatment commenced. The parental PDX was responsive to treatment with fulvestrant and palbociclib alone and in combination. A fresh KCC_P_3837-FPR PDX constituting the third passage subjected to selection was harvested and 4 mm$^3$ sections were implanted into the 4th inguinal mammary gland of 6–8-week-old female NOD-SCID-IL2γR−/− mice (Australian BioResources Pty Ltd). After two weeks to allow for recovery from surgery, mice were randomized to two treatment arms: combined fulvestrant (100 mg/kg in castor oil, once weekly via subcutaneous injection) and palbociclib (25 mg/kg in 2.5% DMSO, 25% β-cyclodextrin, 5 days per week by oral gavage); and triple combination of fulvestrant, palbociclib and capivasertib (100 mg/kg in 2.5% DMSO, 25% β-cyclodextrin, 5 days per week by oral gavage). Tumor growth was supported by implantation of a silastic pellet containing 0.36 mg 17β-estradiol and was monitored visually and by caliper measurement. Endpoint events were tumor width of at least 5 mm (defined as progression) or 60 days of treatment. Procedures and endpoints involving laboratory animals were approved by the Garvan Institute of Medical Research Animal Ethics Committee (protocols 15/25, 18/20, and 18/26).

### Clinical samples and endpoints.
FFPE metastatic lesions from ER+ breast cancer patients treated with combined CDK4/6i and endocrine therapy in the advanced setting were selected retrospectively by database extraction from the archives of the Department of Pathology at Odense University Hospital (OUH) ($N = 115$). The inclusion criteria were ER+ breast cancer patients treated with combined CDK4/6i and endocrine therapy in the advanced setting who had undergone surgery or biopsy for the advanced-stage disease at OUH, and for whom complete clinical information and pathological verification that the metastatic lesion was of breast cancer origin was available. Exclusion criteria were insufficient tumor material in the FFPE block and metastatic biopsy only available after commencing treatment with combined CDK4/6i and endocrine therapy. These parameters yielded $N = 101$ patients. Patients with metastatic biopsies from 2019 ($N = 17$) were included in a pilot cohort that was used to select the cut-off based on the survival significance. Patients with metastatic biopsies obtained before 2019 ($N = 84$) were included in the validation cohort and the cut-off selected in the pilot cohort was applied to stratify patients into p-AKT low and high groups. Tumors were defined ER+ if ≥1% of the tumor cells were stained positive. Progression-free survival (PFS) was defined as the time from initiation of combined endocrine therapy and CDK4/6i treatment until disease progression or death. All clinical samples were coded to maintain patient confidentiality and studies were approved by the Ethics Committee of the Region of Southern Denmark and Copenhagen and Frederiksberg Counties (approval no S-2008-0115). All tissue samples were collected in compliance with the informed consent policy.

### Immunohistochemistry.
FFPE sections (4 μm) of mice tumors and lungs and patients' metastatic lesions were cut with a microtome, mounted on ChemMateTM Capillary Gap Slides (Dako), dried at 60 °C, deparaffinized, and hydrated. Endogenous peroxidase was blocked by 1.5% hydrogen peroxide in TBS buffer, pH 7.4, for 10 min. Antigen retrieval was performed by pretreatment with cell conditioner 1 (CC1) buffer for 32 min at 100 °C or 36 min at 36 °C, or by boiling sections in T-EG solution/TRS buffer (Dako). Primary antibodies used were: anti-Ki67 (790–4286) antibody from Ventana Medical Systems, anti-cleaved caspase-3 antibody (9664) from Cell Signaling Technology, anti-cytokeratin antibody (M351501-2) from Agilent, and anti-pAKT S473 (4060) from Cell Signaling Technology. Primary antibody binding was detected with Optiview-DAB (8-8) for anti-Ki67 and anti-cytokeratin, EnV, FLEX/HRP + Rabbit LINK 15–30 for anti-cleaved caspase-3 and DAB detection kit (Ventana Medical Systems) for anti-pAKT. Sections were also stained with hematoxylin and eosin (HE). Microscopy was performed on a Leica DMLB microscope (×200/numerical aperture (NA) 1.25, Leica Microsystems) using LasV3.6 acquisition software. The Ki67 and cleaved caspase-3 staining was quantified by scanning a representative area of the tumors in each treatment group using ImageJ analysis[55]. Evaluation of the clinical samples was performed by an experienced breast pathologist in a blinded setup. p-AKT expression was observed in the cell nucleus and cytoplasm and tumors were scored based on the H-score calculated by multiplying the percentage of positive tumor

cells (0–100%) by the staining intensity (0–3). The cut-off value for high (H-score ≥ 150) vs. low (H-score < 150) was determined in the pilot cohort based on the survival significance, and the same cut-off was then applied to the validation cohort.

**Statistical analysis**. A two-tailed t-test, ANOVA, or Mann-Whitney-Wilcoxon test were employed for in vitro and in vivo studies (indicated in the figure legends). Grubbs's test was used to find and exclude a single outlier in the dataset. Analysis of growth rate of mice tumors between different treatment groups was performed by a linear mixed-effects model with categorical treatment groups and continuous-time as fixed effects including the interaction. The linear mixed-effects model contains a random effect for the individual tumors to consider repeated measurements within each mouse. The group-specific time slope is referred to as growth rates (GR). For the PDX model, event curves were calculated using the in-built survival analysis, and curves were compared using the log-rank test. For the clinical data, survival curves were generated by Kaplan-Meier estimates by log-rank test to estimate the correlation between p-AKT expression and PFS. Association between p-AKT expression and patient clinicopathological parameters was determined by Fisher's exact and chi-square ($\chi^2$) tests. Cox proportional hazard regression model was used to assess the adjusted hazard ratio (HR) of PFS by p-AKT expression and clinicopathological characteristics. For statistical analysis, STATA v16.0 (STATACorp) and GraphPad Prism v8 (GraphPad Software, Inc.) were used. p values < 0.05 were considered statistically significant.

**Reporting summary**. Further information on research design is available in the Nature Research Reporting Summary linked to this article.

## Data availability

Survival analyses and immunohistochemistry data, are not publicly available to protect patient privacy but will be made available to authorized researchers who have an approved Institutional Review Board application and have obtained approval from The Regional Committees on Health Research Ethics for Southern Denmark. Please contact the corresponding author with data access requests. All other datasets generated during the study are available in the source data file. Source data are provided with this paper.

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

## Acknowledgements

We would like to thank Anne E. Lykkesfeldt for providing MCF-7 and T47D fulvestrant-resistant cell line models, Lisbet Mortensen and Ole Nielsen at the Department of Pathology, Odense University Hospital, for excellent technical assistance with the immunohistochemistry, and M. Kat Occhipinti for editorial assistance. This study was supported by grants from the Danish Cancer Society, A Race Against Breast Cancer, Region of Southern Denmark Research Foundation, and Odense University Hospital Research Council to H.J.D. E.L. is supported by a National Breast Cancer Foundation endowed chair and Love Your Sister.

## Author contributions

Conceptualization: C.L.A., and H.J.D. Methodology: C.L.A., S.E., M.G.T., N.P., M.T., O.L.G., K.K., L.E.J., G.H., A.B., E.L., and H.J.D. Investigation: C.L.A., S.E., M.G.T., N.P., M.T., O.L.G., M.F.H., L.E.J., and M.B. Writing–original draft: C.L.A. and H.J.D. Writing–review, and editing: all authors. Funding acquisition: C.L.A. and H.J.D. Resources: H.J.D. Supervision: H.J.D.

## Competing interests

The authors declare no competing interests.
