## [Peer Review File · Nature Communications]

Reviewers' Comments:

Reviewer #1:

Remarks to the Author:

This study presents findings on the utility of adding the AKT inhibitor capivasertib to the standard treatments of Fulvestrant and Palbociclib. Drug combinations are tested across cell line models of breast cancer grown in vitro or as xenografts.

The addition of a third line of therapy to endocrine therapy/ CDK4-6 inhibitors is currently an intense area of investigation, including in clinical trials. The current manuscript presents data relating to a drug that is clinically well advanced (Capivasertib), and shows some nice data from triplet combinations across cell lines as in vitro and xenograft models. However there are several factors which diminish enthusiasm for this manuscript.

1. Limited and poorly characterised models, particularly in vivo

In vitro:

a. The authors use three well known ER+ models as drug sensitive models, MCF7, T47D and ZR75, and five resistance models are presented: 2 x MCF-7, 1 x ZR-75 and 2 x T-47D. These models are not used consistently throughout the whole manuscript. For example,

- in Figure 1 long term growth assays are not done on ZR-75s as well as the other models.

- in Figure 4I long term growth assays are not performed on T-47D and ZR-75 resistant cell lines, but only on the MCF-7 resistant cell line.

The consequence is that several findings only pertain to 1-2 models, rather than reflecting findings across the whole panel of models.

b. A second major issue with the models is that no raw data of IC50s are shown for the models, and the IC50 values are not specifically identified across the models. This makes it very difficult to compare the models. Notably the ZR-75 cells are treated with a very high dose of 5 micromolar Palbociclib. Is this specific action? Palbociclib inhibits DYRK1A at 2 micromolar doses, which is a proteosomal regulator (Fry et al Mol Cancer Ther 2004 (3) (11) 1427-1438).

c. A third issue is the poor overall characterization of resistance in the models. Do the Palbociclib resistant models show sustained pRB with Palbociclib treatment compared to the sensitive cell lines? Figure 5D is the best data from which to assess this from what is presented, but the data is not presented on single western blots, and is not quantitated. However, from Figure 5D it appears that Palbociclib treatment of the parental MCF7 cell line does not even reduce pRB levels, which raises the question of whether the authors are using appropriate doses and timepoints for Palbociclib treatment (see Point 3 below).

In vivo:

The in vivo data is limited – the experiments are cell line xenografts, and uses cells from only one background, MCF-7. The authors talk about sustained long-term effects of the drug combination, but run the experiments for only 7-8 weeks. The finding that triplet combinations are more effective derives in each case from an analysis of a single timepoint at the conclusion of the experiment. But what happens at Week 9 and 10, is this difference sustained? Is this actually a meaningful and sustained difference that would justify adding extra therapies to a patient's treatment?

Importantly, the authors lack correlative evidence from other endpoints – eg decreased Ki67 in the tumors and increased apoptosis, which they assert are important in their in vitro assays. Finally, these simple xenograft assays fail to address the impact of the triplet therapy on metastasis, which could be achieved by using intraductal xenografting.

2. Companion biomarker doesn't relate to the laboratory models

The authors propose "that patients with metastasis exhibiting high levels of p-AKT are associated with worse prognosis on treatment with standard CDK4/6i and endocrine therapy and may benefit from the addition of an AKTi to improve survival. This does not relate to their in vitro experiments. pAKT levels and Capivasertib sensitivity do not seem to be related. For example, ZR75 cells have relatively little pAKT473, but have the highest sensitivity to Capivasertib as they are treated with the lowest dose of Capivasertib (150nM - which is apparently derived from the IC50 of the cell line, according to the methods).

Also, why do the T-47Ds show decreased pAKT473 in the resistant cell lines? Of the five resistance models presented (2 x MCF-7, 1 x ZR-75 and 2 x T-47D), 2 of the cell lines show pAKT decreases with acquisition of resistance, which is opposite to what is being proposed in the manuscript.

A much more powerful analysis in this context would use cells from Palbociclib resistant patient derived xenografts which have low or high levels of pAKT in their tumors, and then determine their response to the triple line therapy.

3. Datasets lack completeness

Figure 3F – why these are these treatment groups only measured at two weeks? It is clear from the data in figure 3D-3E that the differences may be detectable at later timepoints, and indeed that is the point made by the authors in Figure 3E. They also make points about the duration of response, and this also can't be seen from single timepoints.

Overall the data is not presented in a comprehensive manner that allows for adequate interpretation. For example, in figures 2 & 5 the drug doses and the time of treatment cannot be discerned from the figure or figure legend, and the length of treatment cannot be identified from the methods. How long were the treatments? This is crucial information, as Palbociclib effects are very time dependent, often with different effects at 24h and 72h, as immediate growth suppression becomes released (eg see Herrera-Abreu et al, Cancer Research (2016)). This important concept is not addressed in this manuscript.

4. Lack of mechanism

The authors don't offer an explanation as to why AKT activity increases with Palbociclib resistance, apart from citing the previous paper of Herrera-Abreu et al (2016). There are now multiple CDK4/6 inhibitors on the market including Abemaciclib, which has a different spectrum of target kinases. Without understanding the underlying mechanism behind the increase in AKT activity, it is not possible to know whether these findings will apply to resistance to all CDK4/6 inhibitors.

Other issues:

Figure 2: Apoptosis results are strange. T47Ds are in the range of 0-5 with very small error bars, whereas MCF7s and ZR75s are in the range of 0-160 with very very large error bars. How does the same assay give results of such different quality?

Bar graph data: The data is not presented at a high enough standard to demonstrate transparency about replicate experiments. As stated in <https://www.nature.com/articles/s41467-018-06012-8> bar graphs should show individual data points. In this manuscript, another issue is that each bar graph shows data "relative to vehicle" rather than showing the raw vehicle data and the data points for each treatment in comparison. This doesn't allow the reader to see the natural variability in the data and draw conclusions about the robustness of the dataset.

Figures: are not referred to in order in the text.

Figure 4K: This shows the lysates of three tumors, with a single example of each tumor type. Given the variability of tumor growth in 4J, the authors should show lysates from multiple tumors (eg 3 per cohort) quantitate the western blots, and statistically analyze the data for differences.

Reviewer #2:

Remarks to the Author:

This is a very valuable contribution to the literature.

It has long been known that combined inhibition of PI3K and CDK4/6 is necessary to inhibit CyclinD1. I have treated many patients with the combination of faslodex + evero and palbo (at reduced doses) with great success.

Targeting downstream of PI3K at the level of AKT, combined with CDK4/6 inhibition (+faslodex) is the next much needed step towards curing ER+ve/ Her2-ve breast cancer

This study incorporates 3 cell lines (MCF7, T47D and ZR-75-1), demonstrates the importance of pAKT as a prognostic marker, and demonstrates that use of an AKTi is effective irrespective of PIK3CA and PTEN status.

My only question is why not use upfront, instead of waiting for the patient to progress on first line therapy??

Thank you for your interest in considering a revised version of our manuscript "Co-targeting CDK4/6 and AKT with endocrine therapy prevents progression in CDK4/6 inhibitor and endocrine therapy-resistant breast cancer" by Alves *et al.* We appreciate the insightful comments of the reviewers and have modified the manuscript accordingly.

Below please find a point-by-point response to the reviewer's comments. Revisions in the manuscript are highlighted in red.

Response to comments by Reviewer #1:

This study presents findings on the utility of adding the AKT inhibitor capivasertib to the standard treatments of Fulvestrant and Palbociclib. Drug combinations are tested across cell line models of breast cancer grown in vitro or as xenografts.

The addition of a third line of therapy to endocrine therapy/ CDK4-6 inhibitors is currently an intense area of investigation, including in clinical trials. The current manuscript presents data relating to a drug that is clinically well advanced (Capivasertib), and shows some nice data from triplet combinations across cell lines as in vitro and xenograft models. However there are several factors which diminish enthusiasm for this manuscript.

1. Limited and poorly characterised models, particularly in vivo

In vitro:

a. The authors use three well known ER+ models as drug sensitive models, MCF7, T47D and ZR75, and five resistance models are presented: 2 x MCF-7, 1 x ZR-75 and 2 x T-47D. These models are not used consistently throughout the whole manuscript. For example,

- in Figure 1 long term growth assays are not done on ZR-75s as well as the other models.

Our response: As requested by the reviewer, we have now included a long-term growth assay on ZR-75-1 and ZR-75-1 R. ZR-75-1 R cells showed high sensitivity to AKTi and CDK4/6i single agents when treated for extended periods of time. As mentioned by the reviewer in 1b., the concentration of CDK4/6i used in ZR-75-1 cells was quite high (5 μ M), which could suggest an unspecific effect of CDK4/6i on growth of ZR-75-1 cells. However, lower concentrations of CDK4/6i had almost no effect on growth of ZR-75-1 cells in the short-term experiments (4-6 days) as shown in Fig. 1I and in a new Supplementary Fig. S1 with dose-dependent curves and IC50 (please also see our response to 1b.). Nevertheless, our data suggest that

simultaneous inhibition of ER, CDK4/6 and AKT is required to durably suppress growth of endocrine-resistant cells of different breast cancer cell models. Long-term growth assays in ZR-75-1 cells are included in Supplementary Fig. S3 and Results (p 6).

- in Figure 4I long term growth assays are not performed on T-47D and ZR-75 resistant cell lines, but only on the MCF-7 resistant cell line. The consequence is that several findings only pertain to 1-2 models, rather than reflecting findings across the whole panel of models.

Our response: As requested by the reviewer, we have now included a long-term growth assay on TPF-R cells showing that the triple combination inhibited growth of resistant colonies over the entire 12 weeks of treatment. A double-resistant model derived from ZR-75-1 cells was not included in the manuscript, only from MCF-7 (MPF-R) and T47D (TPF-R) cells. The data from the long-term growth assay on TPF-R cells has been included in Fig. 4J and Results (p 11).

b. A second major issue with the models is that no raw data of IC50s are shown for the models, and the IC50 values are not specifically identified across the models. This makes it very difficult to compare the models. Notably the ZR-75 cells are treated with a very high dose of 5 micromolar Palbociclib. Is this specific action? Palbociclib inhibits DYRK1A at 2 micromolar doses, which is a proteosomal regulator (Fry et al Mol Cancer Ther 2004 (3) (11) 1427-1438).

Our response: As requested by the reviewer, we have now included dose-dependent curves for CDK4/6i and AKTi and the raw data of IC50 of these drugs across the different cell models. The concentration of CDK4/6i and AKTi used in MCF-7, T47D and ZR-75-1 fulvestrant-resistant cell models was determined based on the highest IC50 between parental and resistant cell lines of each cell model. The concentrations of AKTi used in the MPF-R and TPF-R cell models were similar to those used for the MCF-7 and T47D fulvestrant-resistant models and were determined based on the highest IC50 between parental and resistant cell lines of each cell model. These data are included in Supplementary Fig. S1, Results (p 6 and 10) and Methods (p 21).

c. A third issue is the poor overall characterization of resistance in the models. Do the Palbociclib resistant models show sustained pRB with Palbociclib treatment compared to the sensitive cell lines? Figure 5D is the best data from which to assess this from what is presented, but the data is not presented on single western blots, and is not quantitated. However, from Figure 5D it appears that Palbociclib treatment of the parental MCF7 cell line does not even reduce pRB levels, which raises the question of whether the authors are using appropriate doses and timepoints for Palbociclib treatment (see Point 3 below).

Our response: As requested by the reviewer, we have now performed densitometry of p-Rb and Rb bands in Fig. 5D in uncropped full blots of sensitive (M-S and T-S) and resistant (MPF-R and TPF-R) cells exposed together. TPF-R blots in Fig. 5D have been replaced in the revised version of the manuscript to include those exposed together with T-S. We found that Rb and p-Rb S780 (lower band) levels were markedly reduced in M-S and T-S cells after treatment with CDK4/6i alone and combined with fulvestrant, which suggest that the dose and timepoint for CDK4/6i treatment is appropriate. Additionally, we observed that treatment with CDK4/6i alone and combined with fulvestrant decreased Rb levels to a lower extent in MPF-R and TPF-R cells than in the parental cells. Also, p-Rb S780 baseline levels were significantly lower in MPF-R and TPF-R cells compared to the parental cells. Together, these data suggest reduced dependence on cyclin D1-CDK4/6 pathway in MPF-R and TPF-R cells, which is in line with data from Herrera-Abreu et al., Cancer Research (2016). Densitometry data of Rb and p-Rb S780 in Fig. 5D are included in Supplementary Fig. S6 and Results (p 11) and uncropped full blots of Fig. 5D are included in Supplementary Fig. S10.

In vivo:

The in vivo data is limited – the experiments are cell line xenografts, and uses cells from only one background, MCF-7.

Our response: As requested by the reviewer, we have now included cell line xenografts from another background, T47D cells. We have used TPF-R cells to investigate the efficacy of triple therapy on metastasis using an experimental metastasis model (see below). These data are included in Fig. 6D-F and Results (p 13).

The authors talk about sustained long-term effects of the drug combination, but run the experiments for only 7-8 weeks. The finding that triplet combinations are more effective derives in each case from an analysis of a single timepoint at the conclusion of the experiment. But what happens at Week 9 and 10, is this difference sustained? Is this actually a meaningful and sustained difference that would justify adding extra therapies to a patient's treatment?

Our response: We don't believe it is ethically justifiable in accordance with animal welfare to extensively prolong mice experiments with daily treatment. To investigate whether the difference of efficacy between triple and double combination in xenografts is sustained, we have performed statistical analysis with linear mixed effects models to evaluate differences in tumor growth rates (increase or decrease of tumor volume/week) between triple and double combination groups. A statistically significant difference in tumor growth rate between double and triple combinations was observed for both the 182R-1 (GR 7.47, $p = 0.009$, CI 1.88- 13.07) and MPF-R (GR 13.36, $p < 0.0001$, CI 8.41-18.29) xenograft models. These data are included in Results (p 9 and p 13) and Methods (p 25).

Importantly, the authors lack correlative evidence from other endpoints – eg decreased Ki67 in the tumors and increased apoptosis, which they assert are important in their in vitro assays.

Our response: As requested by the reviewer, we have now included analysis of the expression of the proliferation marker Ki67 and the apoptosis marker cleaved caspase-3 by immunohistochemistry in mice tumors. We observed a decrease in Ki67 expression in fulvestrant-resistant (182R-1) tumors and both decreased Ki67 and increased cleaved caspase-3 in fulv+CDK4/6i-resistant (MPF-R) tumors treated with triple combination compared to double combination fulv+CDK4/6i at the endpoint. We believe these data are in line with our vitro results. Data on Ki67 and cleaved caspase-3 in 3 endpoint tumors from each group in Fig. 3E and Fig. 6A are included in Supplementary Figs. S4F and S7B, respectively, Results (p 10 and p 13, respectively) and Methods (p 25).

Finally, these simple xenograft assays fail to address the impact of the triplet therapy on metastasis, which could be achieved by using intraductal xenografting.

Our response: First, we do not believe our xenograft models are simple, as the cells used are either single- or double-resistant. Experimental analysis of metastasis development is a general problem in ER+ breast cancer as none of the existing ER+ BC cell lines metastasize in mice when implanted orthotopically in the fat pad. Intraductal xenografting has been described, but very few labs have mastered the technique. Furthermore, metastases in the lungs, brain and liver will not develop until after 180-270 days post-injection. It is not realistic or ethically justifiable in accordance with animal welfare to perform daily treatments of mice for such a long period. An alternative method is to perform tail-vein injection of cancer cells where most cells end up in the lungs and form tumors there. The model is referred to an experimental metastasis model. Although this model can be criticized for not evaluating all steps of the metastatic process, but only extravasion and colonization, by pre-treating cells in vitro 3 days before injection (to inhibit the signaling pathways without affecting viability) we believe we have mimicked the scenario

wherein tumor cells under selective pressure of the treatment enter the circulation for subsequent development of metastasis in lungs and liver. We have now performed this experimental metastasis model in both MPF-R and TPF-R cells treated for 6 weeks with either the triple combination, fulvestrant+CDK4/6i or fulvestrant+AKTi. We found that the triple combination-treated group contained significantly fewer metastasis (defined, approximately, as tumor area > 2500 μm^2) compared to the combined fulvestrant and CDK4/6i-treated group at 6 weeks of treatment in both MPF-R ($p = 0.050$) and TPF-R ($p = 0.043$) models. For the TPF-R model, the number and size of metastasis was small, likely due to the slow growth rate of these cells. Overall, the metastasis was smaller in the triple combination-treated group compared to the fulvestrant and CDK4/6i combination-treated group in both MPF-R and TPF-R models. Furthermore, the triple combination-treated group showed significantly fewer metastasis than the fulvestrant and AKTi combination-treated group in the MPF-R model ($p = 0.001$). These data are included in Fig. 6D-F, Results (p 13) and Methods (p 23).

2. Companion biomarker doesn't relate to the laboratory models

The authors propose "that patients with metastasis exhibiting high levels of p-AKT are associated with worse prognosis on treatment with standard CDK4/6i and endocrine therapy and may benefit from the addition of an AKTi to improve survival. This does not relate to their in vitro experiments. pAKT levels and Capiwasertib sensitivity do not seem to be related. For example, ZR75 cells have relatively little pAKT473, but have the highest sensitivity to Capiwasertib as they are treated with the lowest dose of Capiwasertib (150nM - which is apparently derived from the IC50 of the cell line, according to the methods). Also, why do the T-47Ds show decreased pAKT473 in the resistant cell lines? Of the five resistance models presented (2 x MCF-7, 1 x ZR-75 and 2 x T-47D), 2 of the cell lines show pAKT decreases with acquisition of resistance, which is opposite to what is being proposed in the manuscript.

Our response: In our original manuscript, we proposed p-AKT as a prognostic marker of outcome of combined CDK4/6i+endocrine treatment, not a marker of AKTi sensitivity. Nevertheless, looking at the uncropped full blots of p-AKT S473 levels across all cell line models now included in Supplementary Figs. S9 and S10, we observed that MCF-7 cells exhibit the lowest p-AKT S473 level and are treated with the highest dose of AKTi (500nM), while ZR-75-1 and T47D cells exhibit higher p-AKT S473 and are treated with lower, and very similar, doses of AKTi (150 nM and 200 nM, respectively). Therefore, it could be argued that p-AKT levels and AKTi sensitivity might be related, which is in line with the data recently published by Gris-Oliver et al., 2020. Finally, we observed that T47D R and TPF-R cells express high levels of p-AKT, even though these are lower than in the parental cells, which indicates that T47D R and TPF-R cells are highly sensitive to AKT inhibition. These data are included in Results (p 8). Uncropped full blots of Western blot bands from Figs. 2 and 5 are included in Supplementary Figs. S9 and S10, respectively. Dose-dependent curves of AKTi in all cell line models are included in Supplementary Fig. S1.

A much more powerful analysis in this context would use cells from Palbociclib resistant patient derived xenografts which have low or high levels of pAKT in their tumors, and then determine their response to the triple line therapy.

Our response: We agree that a PDX model supporting these data would be relevant, however these models are quite difficult to develop and very few groups have successfully done so. The PDX tumor should be from a metastasis that has developed resistance to combined CDK4/6i+endocrine therapy to demonstrate the superiority of the triple versus double combination. Biopsies at this stage are rarely obtained as they do not lead to altered treatment decisions and, even if a biopsy is obtained, the tissue material is generally so small that it is not possible to expand it in mice. Although we have generated several breast cancer PDX models, none fulfill the required conditions. One research group (in Barcelona, Spain) has developed PDX with ER+ tumors resistant to CDK4/6i by exposing the PDX mice to increasing doses of CDK4/6i for months.

However, this model is only resistant to CDK4/6i and not to the combination of CDK4/6i and endocrine therapy, and thus is not relevant. Another research group (in Paris, France) has PDXs that are endocrine-resistant, but not CDK4/6i-resistant, which is not ideal. We have established a collaboration with Dr. Elgene Lim from Garvan Institute of Medical Research, Sydney, who has generated a PDX model that is resistant to both CDK4/6i and endocrine therapy. However, due to the closure of their lab earlier this year as a result of the COVID-19 pandemic and the slow growth rate of ER+ breast cancer PDXs, the expansion of the mouse cohort has been delayed and treatment has just been initiated a few days ago. Therefore, we were not able to include the data in this revised version of the manuscript.

Even more challenging would be development of ER+ PDXs with variable levels of p-AKT S473. Nevertheless, based on our data from in vitro and cell line xenograft models showing high p-AKT in CDK4/6i and fulvestrant-resistant tumors, which respond to triple combination with AKTi, and the data published by Gris-Oliver et al, 2020 showing that PDXs exhibiting high p-AKT levels correlate with higher sensitivity to AKTi, we believe we have provided strong support for the addition of AKTi in ER+ breast cancer patients who progress on CDK4/6i and endocrine therapy.

3. Datasets lack completeness

Figure 3F – why these are these treatment groups only measured at two weeks? It is clear from the data in figure 3D-3E that the differences may be detectable at later timepoints, and indeed that is the point made by the authors in Figure 3E. They also make points about the duration of response, and this also can't be seen from single timepoints.

Our response: The purpose of Fig. 3F is only to show that AKTi alone and Fulv+AKTi does not inhibit tumor growth as effectively as CDK4/6i alone or combinations including CDK4/6i in vivo. As shown in Figs. 3D and 3F, a significant difference in tumor growth between treatment groups can be detected as early as 2 weeks, and therefore we have only included data for these treatments at 2 weeks. Considering the recent findings of the clinical trial FAKTION showing promising results with fulvestrant and AKTi double combination in ER+ advanced breast cancer after previous aromatase inhibitor therapy, we agree that it would be relevant to include a Fulv+AKTi group for longer treatment times. Therefore, we have included a Fulv+AKTi group along with Fulv+CDK4/6i and triple combination groups, all treated for 6 weeks, in the metastasis experimental model now included in the revised manuscript (as mentioned above in point 1).

Overall the data is not presented in a comprehensive manner that allows for adequate interpretation. For example, in figures 2 & 5 the drug doses and the time of treatment cannot be discerned from the figure or figure legend, and the length of treatment cannot be identified from the methods. How long were the treatments? This is crucial information, as Palbociclib effects are very time dependent, often with different effects at 24h and 72h, as immediate growth suppression becomes released (eg see Herrera-Abreu et al, Cancer Research (2016)). This important concept is not addressed in this manuscript.

Our response: As the reviewer rightly points out, drug doses and the time of treatment in Figs. 2 and 5 were missing. We have selected the time point 72h after treatment with drugs to evaluate changes in key molecules of cyclin D/CDK4-6/Rb and PI3K/AKT-mTOR pathways because, as the reviewer mentioned, this is when immediate growth suppression induced by palbociclib is released and early adaptation to CDK4/6 inhibition is observed. We have now included these data in Results (p 7) and legends of Figs. 2 and 5 (p 33 and 39, respectively).

4. Lack of mechanism

The authors don't offer an explanation as to why AKT activity increases with Palbociclib resistance, apart from citing the previous paper of Herrera-Abreu et al (2016). There are now multiple CDK4/6 inhibitors on the market including Abemaciclib, which has a different spectrum of target kinases. Without understanding

the underlying mechanism behind the increase in AKT activity, it is not possible to know whether these findings will apply to resistance to all CDK4/6 inhibitors.

Our response: It is beyond the scope of this manuscript to investigate the molecular mechanism behind high AKT activity in palbociclib resistance. As requested by the reviewer, we have now included growth experiments in both palbociclib-resistant cell line models, MPF-R and TPF-R, with two other CDK4/6i, abemaciclib and ribociclib. We found that both MPF-R and TPF-R show cross-resistance to the other approved CDK4/6i even though these inhibitors exhibit different spectrums of targeted kinases. Furthermore, we observed that the addition of AKTi to the double combination with fulvestrant and ribociclib or abemaciclib induces a greater growth inhibition compared to combined CDK4/6i and fulvestrant. We believe these data suggest that co-targeting CDK4/6 and AKT is efficacious in ER+ breast cancer resistant to the three CDK4/6i currently approved in the clinic. These data are included in Supplementary Fig. S5 and Results (p 11).

Other issues:

Figure 2: Apoptosis results are strange. T47Ds are in the range of 0-5 with very small error bars, whereas MCF7s and ZR75s are in the range of 0-160 with very very large error bars. How does the same assay give results of such different quality?

Our response: We agree with the reviewer that the apoptosis assay shows highly variable results across the cell lines. Importantly, we observed that T47D cells, particularly T47D R and TPF-R, exhibit high baseline (untreated) levels of apoptosis, which is not the case for MCF-7 and ZR-75-1 cells. This is likely the reason why, when presenting the data relative to vehicle, the range of apoptosis values and the error bars are so wide in MCF-7 and ZR-75-1 cells. As the reviewer suggested in the next comment, we have now changed the apoptosis graphs to include individual data points and raw ODs instead of showing data relative to vehicle (Figs. 2A-C and 5A-B), and these changes reduced the range and error bar width variability between the cell lines.

Bar graph data: The data is not presented at a high enough standard to demonstrate transparency about replicate experiments. As stated in <https://www.nature.com/articles/s41467-018-06012-8> bar graphs should show individual data points. In this manuscript, another issue is that each bar graph shows data "relative to vehicle" rather than showing the raw vehicle data and the data points for each treatment in comparison. This doesn't allow the reader to see the natural variability in the data and draw conclusions about the robustness of the dataset.

Our response: As mentioned in the previous comment, we have now changed the bar graphs (cell viability and apoptosis) to include individual data points (Figs. 1, 2, 4 and 5).

Figures: are not referred to in order in the text.

Our response: Figure numbering was changed to address this.

Figure 4K: This shows the lysates of three tumors, with a single example of each tumor type. Given the variability of tumor growth in 4J, the authors should show lysates from multiple tumors (eg 3 per cohort) quantitate the western blots, and statistically analyze the data for differences.

Our response: As requested by the reviewer, we have now included lysates of 3 tumors of each treatment cohort and performed densitometry analysis of Western blot bands. We observed that ER, p-Rb, p-AKT, p-PRAS40 and p-S6 were significantly altered in tumors treated with triple combination, but not in tumors treated with Fulv+CDK4/6i, which support our in vitro data. These data are included in Fig. 6B-C and Results

(p 13).

Response to comments by Reviewer #2:

This is a very valuable contribution to the literature.

It has long been known that combined inhibition of PI3K and CDK4/6 is necessary to inhibit CyclinD1. I have treated many patients with the combination of faslodex + evero and palbo (at reduced doses) with great success.

Targeting downstream of PI3K at the level of AKT, combined with CDK4/6 inhibition (+faslodex) is the next much needed step towards curing ER+ve/ Her2-ve breast cancer

This study incorporates 3 cell lines (MCF7, T47D and ZR-75-1), demonstrates the importance of pAKT as a prognostic marker, and demonstrates that use of an AKTi is effective irrespective of PIK3CA and PTEN status.

My only question is why not use upfront, instead of waiting for the patient to progress on first line therapy??

Our response: We appreciate the positive comments of the reviewer. To answer the reviewer's question, we believe our outgrowth experiments (Fig. 1M-P and Fig. 4I-J) show that although endocrine-resistant and endocrine+CDK4/6i-resistant cells required co-targeting of AKT, CDK4/6i and ER to completely inhibit their growth for 12 weeks course of treatment, growth of endocrine-sensitive cells is effectively inhibited by the clinically-approved double combination fulvestrant+CDK4/6i for the same period of time. Considering the cost and adverse effects associate with the use of triple combination, we believe it is possible to successfully treat many patients with double combination fulvestrant+CDK4/6i, along with the biomarkers for the right selection of patient population that does not required triple combination.

We hope these revisions adequately address the comments of the reviewers and render the manuscript acceptable for publication.

We look forward to your response.

Sincerely yours,

Prof. Henrik Ditzel, MD, PhD, DMSc

University of Southern Denmark

Ph. +4560113781

hditzel@health.sdu.dk

Reviewers' Comments:

Reviewer #2:

Remarks to the Author:

This is an excellent manuscript that provides hard evidence for a triple combination approach, based upon the previous studies that have shown that simultaneous blockade of PI3K and CDK4/6 is needed to completely inhibit Cyclin D1. Also, growth factor-mediated activation of AKT can regulate ER signaling, resulting in ligand-independent activation of the ER genomic pathway. The work is beautifully executed with a massive amount of data in extensive figures, with appropriate cell lines (especially 182R-1 and T47D R), and with appropriate concentrations and controls, both in vitro and in vivo.

Also, the evidence supporting pAKT as a prognostic marker in this population is a substantial contribution to this field.

Nevertheless, as a tumour progresses through each line of therapy, resistance mechanisms accumulate. In second line metastatic disease, it is so difficult to obtain cure. The authors comment that activation of AKT has been associated with resistance to endocrine therapy; hence why treat with only fulvestrant and a CDK 4/6 inhibitor in front line metastatic disease and not the triple combination? This indeed would be a critical contribution.

Reviewer #3:

Remarks to the Author:

In this study, the authors suggest that the AKT pathway is induced under conditions when cells become resistant to CDK4/6 inhibitor based therapy in combination with endocrine treatment. They provide in vitro and in vivo evidence to support their claim that combination of an AKT inhibitor to the CDK4/6i + endocrine therapy would be an effective way to circumvent CDK4/6i resistance. In the first round of the review, the reviewers raised a number of issues with both in vitro and in vivo models and lack of consistency between their models that diminished enthusiasm for the manuscript. The authors attempted to respond to the reviewer's comments by providing additional experiments and analysis. Unfortunately, there are a number of major issues with major inconsistencies across models and across different figures. This was most troubling with the new experiments added in response to reviewers concerns that amplified the issue with inconsistencies across models and that in a number of cases they talk about results that are not shown or over interpreted the results. Lastly, the proposed biomarker and its utility to the treatment strategy do not correlate. As such, the quality of this manuscript is inferior to their original submission and not deemed adequate for publication in Nature Communications. Below is a detailed account of all the issues with this manuscript that we hope will help the authors.

1. In vitro data (Figures 1, 2, 4 and 5)

Figure 1.

- The authors propose that triple combination of CDK4/6i+fulvestrant+AKTi is needed to inhibit the growth of fulvestrant resistant cell lines. However, only 1 (182R-1, panel B) out of 3 cell lines resistant to fulvestrant showed a significant biological difference between fulv+CDK4/6i (blue) and the triple combination (red) with AKTi (panel B vs panels F and J). They also evaluated if the triple combination could delay the emergence of resistant colonies; however, the data from the long-term growth assay was not consistent across all the models (panel M-P and supplemental figure 3). Specifically, in ZR-75-1R cells the combination of fulv+CDK4/6i show the same effect as the triple combination with AKTi, suggesting that addition of AKTi did not further inhibit the growth of cells.

- The combination analysis in supplemental figure S2 was done only in one model (MCF7) and it showed that both strategies, fulv+CDK4/6i or the triple combination have a synergistic effect. Again, no clear advantage to have a triple combination.

- Cell viability was normalized to T=0 instead of vehicle, which is an incorrect way to do the normalization. ZR-75-1 cells did not respond to fulvestrant treatment (panel K), in fact ZR-75-1

cells showed a similar response to resistant cells (ZR-75-1R, panel L). The Y-axis is different in all graphs and makes it difficult to compare across the models. A high concentration of palbociclib (5 μM) was used in ZR-75 cells. The physiological relevant concentration of palbociclib, based on plasma PK in patients is 3.88 μM .

- In response to the reviewers, they included the dose-response curve and the IC50s in the new supplemental figure S1; however, these curves again show inconsistencies, for example, in panel A ZR-75-1R (resistant) cells seems to be more sensitive to CDK4/6i compared to ZR-75-1 parental cells, and compared to the other cell lines. In panel B, the dose-response curves to AKTi did not show a significant difference between resistant cell lines vs parental cell lines. Likewise, the first graph of panel C did not show a significant difference between resistant and parental MCF7 cells to AKTi.
- The authors did not characterize all the cell lines used in a uniform fashion. They needed to show the protein expression of all the pathways studied (additional to the ones showed in fig 2), not just the p-Rb/Rb pathway.

Figure 2.

- Not clear why the basal apoptosis levels are so high in T47D-R (panel B). The author claim that the triple combination is inducing more significant apoptosis in their cell lines, however, their data in Fig 1 is suggestive that there are very modest differences between double and triple combination-hence, not clear that even if apoptosis was in play here why the difference between the double and triple combinations are not more significant.
- The Y axis in panels A ranges are 6 times that of panels B and C, making the comparison between cell lines difficult and also reveals very little difference in apoptosis with the triple combination. Not clear why their error bars are so large in the triple combination in all 3 cell lines.
- Western blot (panel D) also show that T47D R cells have constitutively high levels of cleaved-PARP. This is very different than their other cell lines and suggest that the T47D-R cells may still be undergoing selection. Additionally, levels of the anti-apoptotic protein Bcl-xL was only reduced in ZR-75-1 R cells, the other models did not show any evident changes. Likewise, there is not an evident increase in the levels of the pro-apoptotic protein Bax in any of their models.
- In panel F, there is no consistency in the p-AKT levels amongst their 3 models. No total/p-AKT densitometry is done.
- Panel E also shows inconsistent level of expression amongst their three models. for example, p-AKT levels did not change after triple combination treatment. p-Rb levels also increased after triple combo treatment compared to fulv+CDK4/6i in T47D cells, and they did not show significant changes in T47D R.
- Authors suggest that high p-AKT S473 levels correlate with higher AKTi sensitivity, but these data do not support this claim. Their data completely contradicts this conclusion. Reviewers specifically asked them about this issue; however, it was not addressed. Authors just limited their presentation to include uncropped western blots and tried to provide interpretation of their results that is not supported by the actual data that is presented. In fact, their own data contradicts their interpretation.
- Figures 1 and 2 should have been combined as one figure.
- Additional minor issues with Figure 2 are as follows
 - o No densitometry of any proteins
 - o Inconsistencies between panels E and F
 - o Panels are not cited in order in the text

Figure 4.

- Authors propose that the triple combination is needed to maintain growth inhibition in cells

resistant to both Fulvestrant and CDK4/6i. Only two models are presented and there are inconsistencies between the MPF-R and TPF-R cells. TPF-R cells still respond to the double combination Fulv+CDK4/6i, and in comparison, with the triple combination therapy, there is not a clear and significant difference in the growth curve of their model (panel B vs F).

- Cell viability was normalized to T=0 instead of vehicle, which is incorrect
- Long term growth assays are presented in MCF7 and T47D derived models; however, the data shows inconsistencies. For example, TPF-R cells still responds to CDK4/6i and fulv+CDK4/6i compared to MPF-R cells. Not clear why they did not generate a ZR-75 model resistant to both fulvestrant and CDK4/6i. Huge error bars in their data reduces confidence in their ability to quality control.
- In supplemental figure 5, authors conclude that the cell lines resistant to combined palbociclib and fulvestrant, MPF-R and TPF-R, are cross-resistant to ribociclib and abemaciclib. However, this conclusion is not supported by the data since MPF-R cells exhibit an 82-fold increased palbociclib IC50 (81463 nM) compared to abemaciclib IC50 (992.1 nM), which clearly shows that these cells are not completely cross-resistant to abemaciclib (panel A, right curve). In fact, the single treatment with abema (grey line, panel C) as well as the double combination Fulv+abema (blue line, panel C) reduce the growth of the MPF-R and TPF-R cells to a similar extent than the triple combination (red line, panel C). In contrast, TPF-R are cross-resistant to abemaciclib (panel A). The lack of consistencies across their models for the treatments is suggestive of a potential clonal and not a general effect.

Figure 5.

- Similar to the earlier figures there are numerous inconsistencies across their models. Apoptosis basal levels are particular high in MCF7 models (panel A). Y axis is not uniform and high error bars.
- Western blot (panel C) also shows high cleaved-PARP levels in vehicle treated cells, levels of the anti-apoptotic protein Bcl-xL did not reduce in response to the triple combination treatment; there is not an evident increase in the levels of the pro-apoptotic protein Bax in any model.
- Panel D Lack of consistency in key protein level changes across different models where some models show the desired effect and other models do not. For example, p-AKT levels are reduced in TPF-R cells. Total AKT/ p-AKT needs to be assessed by densitometry.
- Authors suggest that high p-AKT S473 levels correlate with higher AKTi sensitivity, but these data do not support this assumption. Their data completely contradicts this conclusion
- Figures 4 and 5 could have been combined in one single figure.

2. In vivo data (Figures 3 and 6):

Figure 3.

- Authors propose that the addition of AKTi to the standard combination of fulv+CDK4/6i prevents the progression of fulvestrant resistant xenografts. However, the differences in tumor volume after treatment with the triple combination vs the combination of fulv+CDK4/6i are not significant (panel D, red line vs blue line), in fact the tumor growth curves for these conditions are almost identical (differences in tumor weight were also not significant, see Suppl figure 4). Authors argue that when they start the treatment of these tumor at a larger size (panel E), the triple combination is more effective to induce tumor regression than the current therapy fulv+CDK4/6i, this data is controversial regarding the experiment in panel D, and not conclusive since the error bars are large and they overlap one each other.
- In panel E, authors highlight that the combination of fulvestrant and CDK4/6i start to expand after 6 weeks of treatment vs the triple combination, which completely blocked tumor regrowth during 8 weeks. This is hard to asses since the differences in tumor volume were very subtle, the error bars were very large and there is only one measurement for each of the treatment arms. In

addition, the data presented in the supplemental figure S4 at 6 weeks (panel C) vs 8 weeks (panel D) showed that the tumor weight increased at 8 weeks after treatment with either fulv+CDK4/6i or triple combo, which is contradictory with what the authors claim.

- Panel F did not show the tumor volume with key controls: Namely no data of the Fulv+CDK4/6i combination nor the triple combination. In general, is not clear the purpose of this panel, and why the measurements of tumor volume were done at 2 weeks instead at the end-point. Authors did not address this question clearly in their response to the reviewer.
- The initial tumor volume used to start treatment is very different in the vehicle arm across the different panels (B, D, E) in the same figure. This was also observed in the supplemental figure S4 panel B, C, and E.
- The only Xenografts they used is MCF7 parental and resistant cells are showed in this figure, given the lack of consistency of the in vitro data, it would be good to evaluate the effect of these treatments using the other more physiologically relevant models.
- In Supplementary figure S4 (related to figure 3), authors showed a reduction in the expression of Ki67 after the triple combination compared to the double (fulv+CDK4/6i) combination, however they did not present data after treatment with vehicle and single agents, which makes it difficult to assess if the differences are significant. Likewise, the increase in caspase 3 expression is not clear.
- Again, based on these data is not clear if to add an additional therapy to the current standard of care therapy would provide an additional benefit.
- Figure 3 and 6 could have been combined in one single figure

Figure 6.

- In this figure the authors use a cell line that is apparently resistant to CDK4/6i +Fulv however as evident by their own tumor growth curve (Fig 6A), the xenograft model is still responsive to Fulv+Cdk4/6i. Moreover, the differences in tumor volume after treatment with the triple combination vs the combination of fulv+CDK4/6i (panel A, red line vs blue line) are not clear since the error bars overlap (differences in tumor weight are also not clear, see Suppl figure 7A). The double combination of fulv+CDK4/6i should not have induced a significant delay in the tumor growth since the MPF-R cells are resistant to both fulvestrant and CDK4/6i. Same observations for western blots in panel B.
- In Supplementary figure S7 (related to figure 6), authors showed the expression of Ki67 and caspase-3 after the triple combination compared to the double (fulv+CDK4/6i) combination. There is not a visible difference between these treatments. In addition, authors did not present data after treatment with vehicle and single agents, which makes it difficult to assess if the differences are significant.
- Authors included one more cell line xenografts for the metastases experiment, however, similar to the in vitro data there is a lack of consistency between the different models. They observed a nice reduction in the % metastases after triple combination treatment, but only in xenografts derived from MCF7 background. In xenografts from T47D background, the reduction of metastases is observed after both double combination (fulv+CDK4/6i) and triple combination treatment. Besides, the % metastases are very different between both models, TPF-R xenografts showed a small number of metastases compared to MPF-R xenografts.
- Reviewers asked for the inclusion of PDXs models from palbociclib resistant patients with different levels of p-AKT in their tumors in order to determine the response to the triple combination therapy in a more bona-fide model. However, authors did not include any data using the proposed models and only offered a poor justification appealing to the lack of available models with the specific characteristics. There are now patient derived xenograft models that are available from numerous commercial sources that can be used in these studies. Cell line xenografts are not good representation of the clinical experience.

- Panel F did not present data after treatment with vehicle and single agents, which makes difficult asses if the differences are significant

3. Clinical data:

Figure 7.

- Data in figure 7A do not correlate with the in vitro and in vivo data that evaluate the p-AKT levels and the AKTi efficacy/sensitivity.
- Their suggestion that p-AKT can be used as a biomarker is in direct contradiction with their own in vitro and in vivo data which shows that the response to the drug is not dependent on p-AKT
- Lastly, it would have been important for them to show how the AKT inhibitor compares to mTOR or PI3K which are already being used in the clinic.

Ref: NCOMMS-20-07019A

March 4th, 2021

Thank you for your continued interest in considering a revised version of our manuscript "Co-targeting CDK4/6 and AKT with endocrine therapy prevents progression in CDK4/6 inhibitor and endocrine therapy-resistant breast cancer" by Alves *et al.* We appreciate that reviewer #2 commented "The work is beautifully executed with a massive amount of data in extensive figures, with appropriate cell lines (especially 182R-1 and T47D R), and with appropriate concentrations and controls, both in vitro and in vivo. Also, the evidence supporting pAKT as a prognostic marker in this population is a substantial contribution to this field." and had no additional comments. We also appreciate the insightful comments of reviewer #3 and have modified the manuscript accordingly.

Most importantly, we have performed the additional experiments requested by reviewer #3 and incorporated the PDX study with a tumor resistant to combined CDK4/6i and fulvestrant.

Below please find a point-by-point response to reviewer #3 comments. Revisions in the manuscript are highlighted in red.

Response to comments by Reviewer #3:

In this study, the authors suggest that the AKT pathway is induced under conditions when cells become resistant to CDK4/6 inhibitor based therapy in combination with endocrine treatment. They provide in vitro and in vivo evidence to support their claim that combination of an AKT inhibitor to the CDK4/6i + endocrine therapy would be an effective way to circumvent CDK4/6i resistance. In the first round of the review, the reviewers raised a number of issues with both in vitro and in vivo models and lack of consistency between their models that diminished enthusiasm for the manuscript. The authors attempted to respond to the reviewer's comments by providing additional experiments and analysis. Unfortunately, there are a number of major issues with major

inconsistencies across models and across different figures. This was most troubling with the new experiments added in response to reviewers concerns that amplified the issue with inconsistencies across models and that in a number of cases they talk about results that are not shown or over interpreted the results. Lastly, the proposed biomarker and its utility to the treatment strategy do not correlate. As such, the quality of this manuscript is inferior to their original submission and not deemed adequate for publication in Nature Communications. Below is a detailed account of all the issues with this manuscript that we hope will help the authors.

Our response to reviewer's general comments: In the previous revision, reviewer #1 commented about inconsistencies regarding the models used for each experiment throughout the manuscript, meaning that we did not included the same models or all of them (3 fulvestrant-resistant and 2 CDK4/6i+fulvestrant-resistant models) in all experiments. This was optimized in the last revision so that most experiments showed data for all 3/2 models. When reviewer #3 refers to inconsistencies across models he/she is actually referring to the biological variation between the different models. The different ER+ breast cancers cell lines used in our study reflect different cancers that express different levels of signaling pathways regulators, cell replication rate and apoptosis levels. Specifically, the reviewer raises the concern that the different cell models do not show exactly the same expression pattern of protein or apoptosis level, which in fact reflects the biological variation expected in the breast cancer patient population that should receive this treatment. Our findings that the triple combination shows strong growth inhibitory effect and blocks the key signaling pathways in all these diverse fulvestrant-resistant and combined CDK4/6i and fulvestrant-resistant cell models should be considered an advantage and not a limitation.

1. In vitro data (Figures 1, 2, 4 and 5)

Figure 1.

• The authors propose that triple combination of CDK4/6i+fulvestrant+AKTi is needed to inhibit the growth of fulvestrant resistant cells lines. However, only 1 (182R-1, panel B) out of 3 cell lines resistant to fulvestrant showed a significant biological difference between fulv+CDK4/6i (blue) and the triple combination (red) with AKTi (panel B vs panels F and J). They also evaluated if the triple combination could delay the emergence of resistant colonies; however, the data from the long-term growth assay was not consistent across all the models (panel M-P and supplemental figure 3). Specifically, in ZR-75-1R cells the combination of fulv+CDK4/6i show the same effect than the triple combination with AKTi, suggesting that addition of AKTi did not further inhibit the growth of cells.

Our response: We are puzzled that the reviewer claims that a significant biological difference between fulv+CDK4/6i and the triple combination with AKTi was only observed for one (182R-1, panel B) out of 3 fulvestrant-resistant cell lines. We examined this using both crystal violet growth assay (panels A-B, E-F, I-J) and CellTiter-Blue viability assay (panels C-D, G-H, K-L). In fact, we observed statistically significant differences in cell viability between fulv+CDK4/6i and the triple combination in all 3 fulvestrant-resistant cell lines (red vs blue, indicated by single or multiple * ($p < 0.05$) in panels D, H and L). Further, we showed a statistically significant difference in cell growth between fulv+CDK4/6i and the triple combination in 2 of the 3 fulvestrant-resistant cell lines (red vs blue, indicated by single or multiple * ($p < 0.05$) in panels B, F and J). For ZR-75-1 R cells (panel J), reduced cell growth was observed with the triple combination compared to fulv+CDK4/6i, but the difference did not reach significance, likely due to the lower sensitivity of

the growth assay compared to the cell viability assay and the high efficacy of fulv+CDK4/6i in ZR-75-1 R cells. This was also observed in the long-term growth assay, as the reviewer commented, where ZR-75-1 R cells showed high sensitivity to CDK4/6i so no outgrowth was observed. Overall, it can be concluded that the triple combination with AKTi is more efficient than combined fulv+CDK4/6i in most of fulvestrant-resistant cell models, but in some fulvestrant-resistant cells, combined fulv+CDK4/6i is quite effective. We have included this information in Results (p 7).

• *The combination analysis in supplemental figure S2 was done only in one model (MCF7) and it showed that both strategies, fulv+CDK4/6i or the triple combination have a synergistic effect. Again, no clear advantage to have a triple combination.*

Our response: As requested by the reviewer, we have now performed the drug combination analysis in the other two models and included the data in supplementary Figure S2. Stronger synergistic effect, as indicated by lower combination index (CI) value, was observed for the triple combination compared to combined CDK4/6i+Fulv in the fulvestrant-resistant cell lines 182R-1 and T47D R, but not in ZR-75 R. Although the two drug combinations (triple combination and fulv+CDK4/6i) show a synergistic effect, this does not indicate that treatments are equally effective.

• *Cell viability was normalized to T=0 instead of vehicle, which is an incorrect way to do the normalization. ZR-75-1 cells did not respond to fulvestrant treatment (panel K), in fact ZR-75-1 cells showed a similar response to resistant cells (ZR-75-1R, panel L). The Y-axis is different in all graphs and makes it difficult to compare across the models. A high concentration of palbociclib (5 μ M) was used in ZR-75 cells. The physiological relevant concentration of palbociclib, based on plasma PK in patients is 3.88 μ M.*

Our response: As requested by the reviewer, we have now normalized cell viability to vehicle. This did not change the findings of these experiments.

We are puzzled by the reviewer's claims that "ZR-75-1 cells did not respond to fulvestrant treatment (panel K), in fact ZR-75-1 cells showed a similar response to resistant cells (ZR-75-1R, panel L)". ZR-75-1 cells (panel K) showed a significant difference of $p < 0.0001$ (indicated by ***) between vehicle and fulvestrant treatment, while ZR-75-1 R cells (panel L) showed no significant difference (indicated by ns) between vehicle and fulvestrant treatment.

The different range of the Y axis in the graphs is a result of the expected biological variation between cell models.

It is correct that, based on the IC50 titration curves, ZR-75-1 cells seem to be less sensitive to palbociclib than the cells from other models, and therefore a higher concentration of palbociclib (5 μ M) was used for ZR-75-1 cells. This concentration is still within the physiologically-relevant concentration (5 vs 3.88 μ M). We do not find the slightly lower sensitivity of ZR-75-1 cells to palbociclib of concern for the overall conclusion of the study, and we believe this is a result of the normal biological variation observed between different ER+ breast tumors.

• *In response to the reviewers, they included the dose-response curve and the IC50s in the new supplemental figure S1; however, these curves again show inconsistencies, for example, in panel A ZR-75-1R (resistant) cells seems to be more sensitive to CDK4/6i compared to ZR-75-1 parental cells, and compared to the other cell lines. In panel B, the dose-response curves to AKTi did not*

show a significant difference between resistant cell lines vs parental cell lines. Likewise, the first graph of panel C did not show a significant difference between resistant and parental MCF7 cells to AKTi.

Our response: As commented above, what the reviewer refers to as inconsistencies is a result of the biological variation between the breast cancer cell models examined. We agree with the reviewer that ZR-75-1 R cells are more sensitive to CDK4/6i single drug compared to ZR-75-1 parental cells, but ZR-75-1 cells are less sensitive to CDK4/6i (highest IC50) than the other cell lines. This is a reflection of the biological difference between cancer cells, and likely that of clinical specimens, and does not diminish the efficacy of the triple combination in all cell models, including ZR-75-1, which is the main point evaluated in this study. We also agree that the dose-response curves to AKTi did not show a significant difference between resistant cell lines vs parental cell lines in panel B and the first graph of panel C. However, we did not necessarily expect to find a difference in the response to the single AKTi between the resistant and parental model as we are not claiming a benefit of single agent AKTi (which has been previously studied by others), but of the triple combination that includes AKTi in resistant breast cancer.

- The authors did not characterize all the cell lines used in a uniform fashion. They needed to show the protein expression of all the pathways studied (additional to the ones showed in fig 2), not just the p-Rb/Rb pathway.*

Our response: It seems that the reviewer has overlooked the fact that we have also examined the ER and AKT pathways in addition to the p-Rb/Rb pathway in Figure 2E. These are the central pathways inhibited in this study and these analyses were performed uniformly across all 3 fulvestrant-resistant cell lines analyzed (Figure 2E), as well as the combined fulvestrant+CDK4/6i-resistant cell lines (Figure 5D).

- Not clear why the basal apoptosis levels are so high in T47D-R (panel B). The author claim that the triple combination is inducing more significant apoptosis in their cell lines, however, their data in Fig 1 is suggestive that there are very modest differences between double and triple combination- hence, not clear that even if apoptosis was in play here why the difference between the double and triple combinations are not more significant.*

Our response: We agree with the reviewer that it is not clear why the basal apoptosis levels are high in T47D-R, but this has been previously reported (Larsen SL et al., BMC Cancer 2015, 15:239). We consider this a biological variation between the models.

We are puzzled that the reviewer claims that there is only a modest difference in apoptosis between the double (fulv+CDK4/6i) and the triple combination in Fig. 1 (should have been Fig. 2). In all 3 models, we observed a statistically significant difference in apoptosis levels between Fulv+CDK4/6i and the triple combination, as indicated by single or multiple * ($p < 0.05$) in Fig. 2A-C.

- The Y axis in panels A ranges are 6 times that of panels B and C, making the comparison between cell lines difficult and also reveals very little difference in apoptosis with the triple combination. Not clear why their error bars are so large in the triple combination in all 3 cell lines.*

Our response: As mentioned above, the different range of the Y axis between the 3 models (Figure 2A-C) is due to biological differences between cell models. The main point is to compare the efficacy of the different treatments in each model, not to compare a single treatment between the models. Nevertheless, the triple combination consistently induced a significant increase in apoptosis compared to fulv+CDK4/6i in resistant cells of each model (Figure 2A-C). The error bars are large in all 3 cell lines in the triple combination likely due to the high sensitivity of the assay, which can detect differences in apoptosis levels between biological replicates, particularly when the levels are high. To reduce the error bars length, which the reviewer repeatedly comments on, we have changed all graphs to show data as mean \pm SEM instead of SD.

• Western blot (panel D) also show that T47D R cells have constitutively high levels of cleaved-PARP. This is very different than their other cell lines and suggest that the T47D-R cells may still be undergoing selection. Additionally, levels of the anti-apoptotic protein Bcl-xL was only reduced in ZR-75-1 R cells, the other models did not show any evident changes. Likewise, there is not an evident increase in the levels of the pro-apoptotic protein Bax in any of their models.

Our response: We agree with the reviewer that the T47D R cells exhibit constitutively higher levels of cleaved-PARP, which is consistent with the higher basal level of apoptosis observed in these cells, as mentioned above. We also agree with the reviewer's observation that "the anti-apoptotic protein Bcl-xL was only reduced in ZR-75-1 R cells, the other models did not show any evident changes. Likewise, there is not an evident increase in the levels of the pro-apoptotic protein Bax in any of their models". We are not sure why the reviewer highlights this and we don't see this as a concern. This observation only highlights differences in the complex apoptosis pathways between the different models. Importantly, we have performed densitometry of cleaved-PARP bands, which shows that cleaved-PARP is consistently increased following treatment with the triple combination vs. fulv+CDK4/6i in all fulvestrant-resistant cells (panel D).

• In panel F, there is no consistency in the p-AKT levels amongst their 3 models. No total/p-AKT densitometry is done.

Our response: Increased p-AKT levels were observed in resistant cells compared to sensitive cells in 2 (182R-1 and ZR-75-1 R) of the 3 models, as clearly indicated in Results (p 8), but not in T47D R where high p-AKT levels were already observed in the sensitive cell line. Total AKT was not shown in the original Figure 2, but we have now included this analysis (new Fig. 2F and Results p 8). No significant changes were observed in total AKT levels in resistant vs. sensitive cells in the 3 fulvestrant-resistant cell models. As requested by the reviewer, we have also performed densitometry for total/p-AKT (also shown in Fig. 2F).

• Panel E also shows inconsistent level of expression amongst their three models. for example, p-AKT levels did not change after triple combination treatment. p-Rb levels also increased after triple combo treatment compared to fulv+CDK4/6i in T47D cells, and they did not show significant changes in T47D R.

Our response: As already stated in Results (p. 7-8) and shown by others (Ribas, R. et al. Mol.

Cancer Ther. 2015, 14: 2035-2048), treatment with AKTi induces phosphorylation of AKT. No additional increase in p-AKT level was observed or expected after treatment with triple combination for any of the models. It is not clear why the reviewer considers this an inconsistency. We disagree with the reviewer that the p-Rb level did not show significant changes in T47D R cells in triple combination compared to the fulv + CDK4/6i. In all 3 resistant cell lines shown in 2E (182R-1, T47D R and ZR-75-1 R), p-Rb levels are decreased with the triple combination compared to fulv + CDK4/6i, which supports our data that triple combination is the best treatment option in inhibiting growth in resistant cells.

• Authors suggest that high p-AKT S473 levels correlate with higher AKTi sensitivity, but these data do not support this claim. Their data completely contradicts this conclusion. Reviewers specifically asked them about this issue; however, it was not addressed. Authors just limited their presentation to include uncropped western blots and tried to provide interpretation of their results that is not supported by the actual data that is presented. In fact, their own data contradicts their interpretation.

Our response: We suggest that p-AKT S473 levels correlate with response to CDK4/6i+endocrine therapy, and not to AKTi, as stated by the reviewer. This interpretation is based on our preclinical data showing that 3 of 5 cell models of resistance to fulvestrant/fulv+CDK4/6i exhibit high p-AKT compared to the parental (sensitive) cells. The different expression patterns of p-AKT in the remaining 2 models (both derived from T47D cells) might be associated with the very high basal p-AKT level in T47D-sensitive cells. These initial data suggested that p-AKT could be a marker for identification of fulvestrant/fulv+CDK4/6i-resistant tumors, which was further investigated and supported by the clinical data. Since our in vitro and in vivo data showed that the fulvestrant-/fulv+CDK4/6i-resistant cells that exhibit high levels of p-AKT benefit from the addition of AKTi to the standard combination with CDK4/6i and endocrine therapy, we suggest that patients with high-p-AKT tumors are resistant to combined CDK4/6i and endocrine therapy and will likely respond to the triple combination with AKTi. Reviewer #3 and previous reviewer #1 repeatedly comment on a correlation between p-AKT and AKTi sensitivity, but this was not initially addressed in the manuscript. In an attempt to respond to reviewer #1's comment on this, we previously stated in the first response letter that our data on the IC50 of AKTi in the different cell models showed that cells with higher AKTi IC50 (MCF-7) exhibited lower levels of p-AKT, while cells with lower AKTi IC50 (ZR-75-1 and T47D) exhibited higher p-AKT levels, which supported the data previously published by Gris Oliver et al. 2020 showing that high p-AKT S473 correlate with higher AKTi sensitivity. Importantly, the correlation between p-AKT S473 and AKTi sensitivity is not a novel finding in this manuscript and this is clearly stated in Results (p 8).

• Figures 1 and 2 should have been combined as one figure.

Our response: We considered combining Figures 1 and 2, but as these figures contain 16 and 6 subfigures, respectively, we find it impossible to display these at a sufficient size to meet the requirements from the journal, even if some of the subfigures were moved to the Supplemental data. Further, our manuscript includes a total of 7 figures, which is less than most papers published in Nature Communications.

• Additional minor issues with Figure 2 are as follows

- o No densitometry of any proteins*
- o Inconsistencies between panels E and F*
- o Panels are not cited in order in the text*

Our response:

- o Densitometry is now provided.*
- o In panel 2F, no treatment is included so the reviewer should compare p-AKT and pan AKT after treatment with vehicle in Fig 2E (first lane for each cell line) with 2F. The same pattern of band intensities is observed for these proteins in 2E and F.*
- o The order of the panels has been double checked and panels were cited in the correct order in the text.*

Figure 4.

- Authors propose that the triple combination is needed to maintain growth inhibition in cells resistant to both Fulvestrant and CDK4/6i. Only two models are presented and there are inconsistencies between the MPF-R and TPF-R cells. TPF-R cells still respond to the double combination Fulv+CDK4/6i, and in comparison, with the triple combination therapy, there is not a clear and significant difference in the growth curve of their model (panel B vs F).*

Our response: We are puzzled by the reviewer's claims that "TPF-R cells still respond to the double combination Fulv+CDK4/6i, and in comparison, with the triple combination therapy, there is not a clear and significant difference in the growth curve of their model (panel B vs F)." Panels B and F, which correspond to the cell growth assay, clearly show that for both MPF-R and TPF-R cells, no significant difference was observed after treatment with Fulv+CDK4/6i compared to vehicle (black vs blue lines), as indicated by ns, while there was a significant difference in growth with the triple combination compared to Fulv+CDK4/6i (red vs blue lines) as indicated by * $p < 0.05$. The same findings were observed when evaluating cell viability in MPR-R and TPF-R cells (panels D and H). Thus, there are no inconsistencies between the two models.

- Cell viability was normalized to $T=0$ instead of vehicle, which is incorrect*

Our response: As requested by the reviewer, we have now normalized cell viability to vehicle. This did not change the findings of these experiments.

- Long term growth assays are presented in MCF7 and T47D derived models; however, the data shows inconsistencies. For example, TPF-R cells still responds to CDK4/6i and fulv+CDK4/6i compared to MPF-R cells. Not clear why they did not generate a ZR-75 model resistant to both fulvestrant and CDK4/6i. Huge error bars in their data reduces confidence in their ability to quality control.*

Our response: We agree with the reviewer that outgrowth of TPF-R resistant colonies after treatment with CDK4/6i and fulv+CDK4/6i is only observed at week 3 compared to week 1 in MPF-R cells. However, this is not an important point of this study and simply reflects the biological variation between the two models. The important point in the long-term growth assays is that only the triple combination completely suppressed outgrowth of resistant colonies of both MPF-R and TPF-R cells during the entire period of the study (12 weeks). Thus, there are no inconsistencies between the two models. The generation of double resistant cell lines is an enormous amount of

work and two cell models are generally sufficient to validate in vitro findings, as also discussed with the editor, especially when these are supported by in vivo and clinical data.

• In supplemental figure 5, authors conclude that the cell lines resistant to combined palbociclib and fulvestrant, MPF-R and TPF-R, are cross-resistant to ribociclib and abemaciclib. However, this conclusion is not supported by the data since MPF-R cells exhibit an 82-fold increased palbociclib IC50 (81463 nM) compared to abemaciclib IC50 (992.1 nM), which clearly shows that these cells are not completely cross-resistant to abemaciclib (panel A, right curve). In fact, the single treatment with abema (grey line, panel C) as well as the double combination Fulv+abema (blue line, panel C) reduce the growth of the MPF-R and TPF-R cells to a similar extent than the triple combination (red line, panel C). In contrast, TPF-R are cross-resistant to abemaciclib (panel A). The lack of consistencies across their models for the treatments is suggestive of a potential clonal and not a general effect.

Our response: We agree with the reviewer that MPF-R cells are not completely cross-resistant to abemaciclib since the IC50 of abemaciclib is significantly lower than that of palbociclib. Nevertheless, MPF-R cells exhibit a significantly higher (13.5-fold) IC50 of abemaciclib than M-S cells (panel A, right curve), and thus MPF-R are significantly more resistant to abemaciclib than M-S cells. We have changed the wording in the Results (p 11) and Supplementary figure legends to make this clear. On the other hand, we disagree with the reviewer that “In fact, the single treatment with abema (grey line, panel C) as well as the double combination Fulv+abema (blue line, panel C) reduce the growth of the MPF-R and TPF-R cells to a similar extent than the triple combination (red line, panel C)” since the triple combination clearly inhibits growth more than the single agent abemaciclib and combined fulvestrant and abemaciclib in both MPF-R and TPF-R cells, even though the difference is not statistically significant. Further, there is no lack of consistency across the two models, as the reviewer states, since the triple combination with different CDK4/6i is the most effective treatment in both models.

Figure 5.

• Similar to the earlier figures there are numerous inconsistencies across their models. Apoptosis basal levels are particular high in MCF7 models (panel A). Y axis is not uniform and high error bars.

Our response: As mentioned above regarding the apoptosis data in Figure 2, biological differences in apoptosis basal levels between the two models are not inconsistencies and reflect the biological differences that may be observed among cancer patients. The reviewer states that the “apoptosis basal levels are particular high in MCF7 models (panel A)”, but this is not correct. The apoptosis basal level is higher in TPF-R cells from the T47D model (panel B), consistent with what is observed in figure 2 for T47D-R cells. Y axis range is not uniform in the 2 models as a result of the biological variation between the models. The error bars are large due to the high sensitivity of the assay, which is able to detect differences in apoptosis levels between biological replicates, particularly when the levels are high.

• Western blot (panel C) also shows high cleaved-PARP levels in vehicle treated cells, levels of the anti-apoptotic protein Bcl-xL did not reduce in response to the triple combination treatment; there is not an evident increase in the levels of the pro-apoptotic protein Bax in any model.

Our response: Similar to our response above regarding Figure 2, we are not sure why the reviewer

highlights “levels of the anti-apoptotic protein Bcl-xL did not reduce in response to the triple combination treatment; there is not an evident increase in the levels of the pro-apoptotic protein Bax in any model”. This observation only highlights differences in the complex apoptosis pathways between the different models. Importantly, we have performed densitometry of cleaved-PARP bands that shows that cleaved-PARP is consistently increased following treatment with the triple combination vs. fulv+CDK4/6i in both cell models resistant to combined fulvestrant and CDK4/6i (panel C).

- *Panel D Lack of consistency in key protein level changes across different models where some models show the desired effect and other models do not. For example, p-AKT levels are reduced in TPF-R cells. Total AKT/p-AKT needs to be assessed by densitometry.*

Our response: There are no inconsistencies in key protein level changes across the 2 models in panel D. p-Rb, Rb, p-PRAS40 and p-S6 levels are reduced after treatment with triple combination compared to fulv+CDK4/6i in both models. As mentioned before, p-AKT levels are slightly lower in TPF-R than T-S cells, which is the opposite of MPF-R and M-S cells. We believe this might be due to the very high basal p-AKT level in T47D model. Total AKT/p-AKT has now been assessed by densitometry and the data is presented in panel E.

- *Authors suggest that high p-AKT S473 levels correlate with higher AKTi sensitivity, but these data do not support this assumption. Their data completely contradicts this conclusion*

Our response: The data in Figure 5 (panel E) show that p-AKT S473 levels are higher in MPF-R cells, but not in TPF-R cells, compared to the parental cells, and thus we suggest that p-AKT S473 levels may be associated with resistance to combined fulvestrant and CDK4/6i, and not with AKTi sensitivity, as the reviewer states. We further investigated the correlation between p-AKT levels and response to combined CDK4/6i and endocrine therapy in patient tumor biopsies (Figure 7) and found significantly lower progression-free survival in patients with tumors with high p-AKT compared with the low p-AKT tumors, which supports the prognostic value of p-AKT levels in patients treated with combined CDK4/6i and endocrine therapy.

- *Figures 4 and 5 could have been combined in one single figure.*

Our response: We considered combining Figures 4 and 5, but as these figures contain 10 and 5 subfigures, respectively, we find it impossible to display these at a sufficient size to meet the requirements from the journal. Further, our manuscript includes a total of 7 figures, which is less than most papers published in Nature Communications.

2. In vivo data (Figures 3 and 6):

Figure 3.

- *Authors propose that the addition of AKTi to the standard combination of fulv+CDK4/6i prevents the progression of fulvestrant resistant xenografts. However, the differences in tumor volume after treatment with the triple combination vs the combination of fulv+CDK4/6i are not significant (panel D, red line vs blue line), in fact the tumor growth curves for these conditions are almost identical (differences in tumor weight were also not significant, see Suppl figure 4). Authors argue that when they start the treatment of these tumor at a larger size (panel E), the triple combination is more effective to induce tumor regression than the current therapy fulv+CDK4/6i, this data is*

controversial regarding the experiment in panel D, and not conclusive since the error bars are large and they overlap one each other.

Our response: We agree with the reviewer that there are no significant differences in fulvestrant-resistant tumors volume after treatment with the triple combination vs the combination of fulv+CDK4/6i in panel D (red line vs blue line). This is because both drug combinations completely suppress outgrowth of the tumors when they are small at treatment initiation, and we do not claim otherwise in the Results (p 9). In contrast, when the tumors were larger at treatment initiation, the triple combination induced a statistically significant growth inhibition compared to the combination of fulv+CDK4/6i, as analyzed by two-tailed t-test at the endpoint ($p < 0.05$). We do not agree with the reviewer that “this data is not conclusive since the error bars are large and they overlap one each other” as these are already considered in the statistical analysis, and the error bars are not particularly large for an animal experiment. We have also included the non-parametric Wilcoxon test in case the reviewer is concerned whether equal variance and normal distribution between the two groups can be considered. The Wilcoxon test also showed a significant difference in tumor volume between the two treatment groups ($p = 0.009$, Fig. 3E). Further, the reviewer seems to have overlooked the fact that we also performed a statistical analysis evaluating whether the growth curves of the two treatments were different over time. Statistically significant differences in the growth rates between the two treatment arms were observed using linear mixed effects models (GR 7.47, $p = 0.009$, CI 1.88-13.07).

• In panel E, authors highlight that the combination of fulvestrant and CDK4/6i start to expand after 6 weeks of treatment vs the triple combination, which completely blocked tumor regrowth during 8 weeks. This is hard to assess since the differences in tumor volume were very subtle, the error bars were very large and there is only one measurement for each of the treatment arms. In addition, the data presented in the supplemental figure S4 at 6 weeks (panel C) vs 8 weeks (panel D) showed that the tumor weight increased at 8 weeks after treatment with either fulv+CDK4/6i or triple combo, which is contradictory with what the authors claim.

Our response: As mentioned above, the statistical analysis used consider the variability of tumor sizes and demonstrate a statistically significant difference between the two treatments. The reviewer does not indicate that we have used incorrect statistics, and thus we hope that the reviewer would respect the results of the statistical analysis. We are puzzled why the reviewer claims that “the data presented in the supplemental figure S4 at 6 weeks (panel C) vs 8 weeks (panel D) showed that the tumor weight increased at 8 weeks after treatment with either fulv+CDK4/6i or triple combo, which is contradictory with what the authors claim”. Panel C in Figure S4 is related to panel D in Figure 3, where treatment was initiated with smaller tumors, while panel D in Figure S4 is related to panel E in Figure 3, where treatment was initiated when the tumors were larger, as described in the Results (p 9). Therefore, these data are not contradictory and fit perfectly with the tumor volume results in Figure 3 D and E (tumor volume is higher in panel E than in D at the endpoint).

• Panel F did not show the tumor volume with key controls: Namely no data of the Fulv+CDK4/6i combination nor the triple combination. In general, is not clear the purpose of this panel, and why the measurements of tumor volume were done at 2 weeks instead at the end-point. Authors did not address this question clearly in their response to the reviewer.

Our response: As already explained in our previous response letter, the purpose of this experiment was to evaluate whether fulvestrant or AKTi monotherapy or combined fulvestrant and AKTi

significantly inhibited tumor growth, which was not the case. As this is a minor finding, we have moved panel F to Supplementary Figure S4E.

- *The initial tumor volume used to start treatment is very different in the vehicle arm across the different panels (B, D, E) in the same figure. This was also observed in the supplemental figure S4 panel B, C, and E.*

Our response: The initial tumor volume used when starting treatment was approximately the same for B and D but, as already explained in Results (p 9) and above, the tumor volume was intentionally increased in E to more clearly observe a difference in tumor growth inhibition between the triple combination and Fulv+CDK4/6i treatment.

- *The only Xenografts they used is MCF7 parental and resistant cells are showed in this figure, given the lack of consistency of the in vitro data, it would be good to evaluate the effect of these treatments using the other more physiologically relevant models.*

Our response: The main focus of the study is evaluating the efficacy of triple combination with AKT inhibitor in tumors resistant to combined fulvestrant and CDK4/6i, and for this purpose we used 4 different in vivo models, including a PDX model, 2 metastasis models and an orthotopic model. The treatment of fulvestrant-resistant cells is of less importance, which is the reason for including only one in vivo model, which was evaluated in several separate animal experiments.

- *In Supplementary figure S4 (related to figure 3), authors showed a reduction in the expression of Ki67 after the triple combination compared to the double (fulv+CDK4/6i) combination, however they did not present data after treatment with vehicle and single agents, which makes it difficult to assess if the differences are significant. Likewise, the increase in caspase 3 expression is not clear.*

- *Again, based on these data is not clear if to add an additional therapy to the current standard of care therapy would provide an additional benefit.*

Our response: As mentioned above, the main point of these data is to determine whether adding an additional drug (AKTi) to the current standard of care therapy (fulv+CDK4/6i) provides an additional benefit when cancer cells are resistant to either combined fulv+CDK4/6i or fulvestrant alone. Comparison with single agents and vehicle are not critical. Nevertheless, we have now included hematoxylin and eosin (HE) staining of 182R-1 tumors treated for 6 weeks with vehicle, fulvestrant alone, fulvestrant+CDK4/6i and triple combination with AKTi (Supplementary Fig. S4G, Results p 9). These show that tumors treated with vehicle or fulvestrant predominantly consisted of vital tumor cells. Tumors treated with combined CDK4/6i and fulvestrant were smaller and contained infiltrating fat cells. This was much more pronounced in tumors treated with the triple combination where only smaller tumor islets containing central degeneration surrounded by fat tissue was observed. We have also included quantification of Ki67 and cleaved-caspase 3 stainings in Supplementary figure S4H, which shows that tumors treated with the triple combination exhibited a statistically significant reduction of Ki67 expression compared to tumors treated with the double (fulv+CDK4/6i) combination. No difference was observed in cleaved caspase-3 expression between treatments, and thus we revised the Results section accordingly (p 10). To further address the reviewer's concern, we also evaluated the morphology of the 182R-1 tumors from Figure 3E by HE staining. These showed that tumors treated with the triple combination contained large areas of degeneration and reactive fibrosis (within the indicated circles) surrounded

by vital tumor tissue. In contrast, only small areas of degeneration and reactive fibrosis were observed in tumors treated with combined CDK4/6i and fulvestrant and the vital tumor tissue areas were much larger in these tumors (Supplementary Figure S4H, Results p 10). Together, these data support our claim that addition of AKTi to current standard of care therapy provides additional benefit.

- *Figure 3 and 6 could have been combined in one single figure*

Our response: Figure 3 is related to the fulvestrant-resistant mouse model, while Figure 6 is related to the combined CDK4/6i and fulvestrant-resistant mouse and PDX models, thus we do not believe these figures should be combined. As mentioned before, our manuscript only includes a total of 7 figures, which is less than most papers published in Nature Communications, so we do not see a need for reduction.

Figure 6.

- *In this figure the authors use a cell line that is apparently resistant to CDK4/6i +Fulv however as evident by their own tumor growth curve (Fig 6A), the xenograft model is still responsive to Fulv+Cdk4/6i. Moreover, the differences in tumor volume after treatment with the triple combination vs the combination of fulv+CDK4/6i (panel A, red line vs blue line) are not clear since the error bars overlap (differences in tumor weight are also not clear, see Suppl figure 7A). The double combination of fulv+CDK4/6i should not have induced a significant delay in the tumor growth since the MPF-R cells are resistant to both fulvestrant and CDK4/6i. Same observations for western blots in panel B.*

Our response: It is widely known by all who develop and work with drug-resistant cancer cell lines that although they may expand in the presence of the drug they are resistant to, they nearly always show some response to the drug when compared to untreated cells. This behavior correlates to the clinical situation where tumor growth may be slowed by a drug, but if the tumor keeps expanding, it is considered resistant to the drug treatment.

We do not agree with the reviewer that “the differences in tumor volume after treatment with the triple combination vs the combination of fulv+CDK4/6i (panel A, red line vs blue line) are not clear since the error bars overlap (differences in tumor weight are also not clear, see Suppl figure 7A).” Statistical analysis using two-tailed t-test at the endpoint clearly showed a statistically significant difference in tumor volume (Fig. 6A) and tumor weight (Supplementary Fig. S7A) between the two treatments ($p < 0.05$). We have also included the non-parametric Wilcoxon test that showed a significant difference in tumor volume between the two treatment groups ($p = 0.012$, Fig. 6A). Further, a statistically significant difference in tumor growth rate was observed between double and triple combination using linear mixed effects models (GR 13.36, $p < 0.0001$, CI 8.41-18.29).

- *In Supplementary figure S7 (related to figure 6), authors showed the expression of Ki67 and caspase-3 after the triple combination compared to the double (fulv+CDK4/6i) combination. There is not a visible difference between these treatments. In addition, authors did not present data after treatment with vehicle and single agents, which makes it difficult to assess if the differences are significant.*

Our response: As mentioned above regarding Supplementary Figure S4, the main point of these data is to determine whether adding an additional drug (AKTi) to the current standard of care

therapy (fulv+CDK4/6i) provides an additional benefit in tumors resistant to either combined fulv+CDK4/6i or fulvestrant alone. Comparison with single agents and vehicle are not critical. We have included quantification of Ki67 and cleaved-caspase 3 stainings in Supplementary Figure S7C, which shows that tumors treated with the triple combination exhibit a statistically significant reduction of Ki67 expression compared to tumors treated with the double (fulv+CDK4/6i) combination. No difference is observed in cleaved caspase-3 expression between treatments and thus we revised the Results section accordingly (p 13). We also evaluated the morphology by HE staining, which showed that tumors treated with combined CDK4/6i and fulvestrant primarily consisted of vital tumor tissue, while tumors treated with the triple combination were smaller and contained larger areas of degeneration and reactive fibrosis (within the indicated circles) or smaller areas of degeneration and lipid infiltration surrounded by vital tumor tissue (Supplementary Fig. S7C).

• Authors included one more cell line xenografts for the metastases experiment, however, similar to the in vitro data there is a lack of consistency between the different models. They observed a nice reduction in the % metastases after triple combination treatment, but only in xenografts derived from MCF7 background. In xenografts from T47D background, the reduction of metastases is observed after both double combination (fulv+CDK4/6i) and triple combination treatment. Besides, the % metastases are very different between both models, TPF-R xenografts showed a small number of metastases compared to MPF-R xenografts.

Our response: As mentioned several times previously, the MCF7 and T47D-derived models exhibit differences in biology, as would be expected for different ER+ breast tumors in patients. These are not inconsistencies, but biological variations. Such biological variations are also observed in the metastatic potential of the two cell models, where MPF-R cells are more metastatic than TPF-R cells. We disagree with the reviewer that “They observed a nice reduction in the % metastases after triple combination treatment, but only in xenografts derived from MCF7 background. In xenografts from T47D background, the reduction of metastases is observed after both double combination (fulv+CDK4/6i) and triple combination treatment”. Although we observed a reduction in the % of metastasis with fulv+AKTi compared to fulv+CDK4/6i in T47D model, a significant reduction in the % metastases with triple combination treatment compared to the standard double combination (fulv+CDK4/6i) was observed in both MPF-R and TPF-R. These data show again the benefit of the triple combination with AKTi over the standard double combination fulv+CDK4/6i.

• Reviewers asked for the inclusion of PDXs models from palbociclib resistant patients with different levels of p-AKT in their tumors in order to determine the response to the triple combination therapy in a more bona-fide model. However, authors did not include any data using the proposed models and only offered a poor justification appealing to the lack of available models with the specific characteristics. There are now patient derived xenograft models that are available from numerous commercial sources that can be used in these studies. Cell line xenografts are not good representation of the clinical experience.

Our response: We have now included data from a PDX model resistant to combined CDK4/6i and fulvestrant (Figure 6C-D, Results p 13). The data demonstrate that in this model the triple combination also caused a statistically significant delay in the outgrowth of the PDX tumors compared to treatment with combined CDK4/6i and fulvestrant. This PDX was obtained through our new collaborator Prof. Elgene Lim at Galvan Institute of Medical Research, Australia. We have searched all the major commercial sources for PDX models, but an ER+ breast cancer model resistant

to combined CDK4/6i and fulvestrant with different levels of p-AKT is not available. We also evaluated the excised PDX tumors, as the editor proposed. Reduced Ki67 staining was observed in tumors treated with the triple combination compared with the combined CDK4/6i and fulvestrant (Supplementary Figure S7D and Results p 14). Furthermore, HE staining shows that PDX tumors treated with the triple combination contained large areas of degeneration and reactive fibrosis (within the indicated circles) surrounded by vital tumor tissue. In contrast, only small areas, if any, of degeneration and reactive fibrosis were observed in tumors treated with combined CDK4/6i and fulvestrant and the vital tumor tissue area was much larger in these tumors (Supplementary Figure S7D and Results p 14). As the read out of the PDX model was progression-free survival (PFS), with progression defined as tumor width ≥ 5 mm, the tumors receiving the triple combination had started growing (had become resistant to the triple drug combination) before the animals were sacrificed, and thus the inhibition on the signaling pathways as evaluated in our in vitro and other vivo models, was not possible in the PDX tumors.

- *Panel F did not present data after treatment with vehicle and single agents, which makes difficult asses if the differences are significant*

There is no indication from the other in vivo experiments that any of the single drugs are more effective than the combined CDK4/6i and fulvestrant, which is the critical “control group” for comparison with the triple combination since these cells are resistant to combined fulv+CDK4/6i. There is no clear reason to include 40 additional mice (10 mice per each group, 4 additional groups) in a in vivo experiment that is already large.

3. Clinical data:

Figure 7.

- *Data in figure 7A do not correlate with the in vitro and in vivo data that evaluate the p-AKT levels and the AKTi efficacy/sensitivity.*

Our response: The data in figure 7A evaluate the correlation between p-AKT levels and CDK4/6i+endocrine therapy efficacy, and not AKTi efficacy/sensitivity, as the reviewer incorrectly states.

- *Their suggestion that p-AKT can be used as a biomarker is in direct contradiction with their own in vitro and in vivo data which shows that the response to the drug is not dependent on p-AKT*

Our response: As mentioned before, our preclinical data show that 3 of 5 cell models of resistance to fulvestrant/fulv+CDK4/6i exhibit high p-AKT compared to the parental sensitive cells. The different expression patterns of p-AKT in the 2 models derived from T47D cells might be associated with the very high basal p-AKT level in T47D-sensitive cells. As mentioned in the Results and Discussion, these data suggested that p-AKT could be a marker for identification of fulvestrant/fulv+CDK4/6i resistant tumors, which was further investigated and supported by the clinical data. Since our in vitro and in vivo data show that the fulvestrant/fulv+CDK4/6i cells exhibiting high levels of p-AKT benefit from the addition of AKTi to the standard combination with CDK4/6i and endocrine therapy, we suggest that high-p-AKT tumors are resistant to combined CDK4/6i and endocrine therapy and will likely respond to the triple combination with AKTi.

• *Lastly, it would have been important for them to show how the AKT inhibitor compares to mTOR or PI3K which are already being used in the clinic.*

Our response: We agree with the reviewer that comparing the efficacy of triple combinations, including an AKTi or an inhibitor of other regulator of the PI3K pathway such as a mTOR or PI3K inhibitor, would be interesting, but this is beyond the scope of this study.

We hope these revisions adequately address the comments of reviewer #3 and render the manuscript acceptable for publication.

We look forward to your response.

Sincerely yours,

Prof. Henrik Ditzel, MD, PhD, DMSc
University of Southern Denmark
Ph. +4560113781
hditzel@health.sdu.dk

Reviewers' Comments:

Reviewer #3:

Remarks to the Author:

General comments: While the authors performed additional experiments in response to this reviewer, in general the authors did not address the major concerns raised. Of note, lack of consistency amongst different model systems, lack of evidence for why using triple combination is better than two, when both combinations are synergistic, lack of normalization of their data to show true effects and lack of proper controls for their in vivo model systems. The authors still did not include the controls requested for the in vivo study and just presented the H&E for the controls but not the actual staining requested (CC3, Ki67). The results of the single agents should be compared to the double combination vs triple combination. Lastly, the new data in figure 6 again lack of controls. Lastly, it is not clear which is the novelty of this work since currently there is an ongoing clinical trial (CAPITello-292).

Below are a point-by-point to the main issues that have remained unanswered by the authors. Since biological variation is expected in breast cancer patients that potentially will receive a new drug or combined therapy, in vitro studies should provide **STRONG AND CONSISTENT** data over different models that represent such biological variation in order to be sure that the results seen in an experiment are the result of a treatment-related "signal" and not biological "noise". When there is too much biological variation in in vitro studies where most of the conditions can be controlled, the experimenter cannot be sure that the treatment did have an effect, statistical analyses do not give a definite yes or no answer, rather a probability statement. The use of diverse models is the way to include in in vitro studies the expected biological variation. The lack of consistency in response among the different in vitro models used, decreases confidence in the utility of the triple combination.

In vitro data (Figures 1, 2, 4 and 5)

Figure 1. The differences the authors show while statistically significant are unlikely to be biologically significant. The reasons are as follows: taking in consideration the results for the growth assay, cell viability, and the long-term growth assay, only one out of 3 models (182R-1) showed an evident effect. A second model (ZR-75-1R) showed no differences comparing controls to double combination in 2 out of 3 assays, specifically in the long-term growth assay, except for vehicle, all treatments were equally effective. The third model, T47DR, although the differences were statistically significant, the effect seems not very clear in the growth assay and long-term growth assay, since in panel P, T47DR cells are still responsive to fulv+cdk4/6i.

Combination index: Authors included new data with the other models, however, the main concern remains, and it is even more evident since in all the 3 models the double and triple combination as well as the other combinations tested showed to be synergistic. Although, as the authors discussed, this does not suggest that treatments are equally effective, these new results reduce the enthusiasm for the triple combination and raise the question about the advantage to add an additional therapy, which could cause additional adverse events.

Normalization: To properly normalize the data, the authors need to use the same Y axis from model to model, not different Y axis to amplify their effects. Additional dose ranges to use: Not clear why the authors did not compare the more physiological dose of palbo in the ZR751 line.

Uniform characterization of the cell lines: The authors did not characterize all the cell lines used in a uniform fashion. They needed to show the protein expression of all the pathways studied (additional to the ones showed in fig 2), not just the p-Rb/Rb pathway. It seems that the authors are cherry picking which proteins to examine in what cell lines and for what combination. For example, since the authors are proposing triple combination, it would be good to also CDK4, CDK6, Cyclin D, etc;

Results are not consistent with the claims: Authors state on page 8 line 174-175 that high levels of p-akt correlates with high sensitivity to AKT inhibitor; however, there is no differences between double combo fulv+cdk4-6 and the triple combination. Again, only 2 models (182R1 and ZR751R) show that p-AKT473 increase after triple combination compared to fulv+cdk4-6, but this is not in agreement with the results in figure 1. Only the result for 182R1 correlates (high p-akt, high

sensitivity to triple combo). ZR751R show high p-akt but was not sensitive to the triple combination.

Additional minor issues with Figure 2 are as follows

- o No densitometry for panel E protein.
- o Inconsistencies between panels E and F: p-aktS473 is high for 182R1 cells in panel F but very low in panel E (vehicle), same thing for T47DR, ZR751 and ZR751R

Figure 4.

Panel F-it is curious that the differences between black and blue lines is not statistically significant, but the differences between blue and red (which is similar to the black and blue, around 2 times) is statistically significant. A more rigorous review of the statistical analysis is required.

The outgrowth of TPFR resistant colonies treated with CDK4/6i start the log phase after week 7, which suggest that these cells are still responsive and are not fully resistant as the authors state. It is an important point because, again, it raises the question about the advantage to add an additional therapy (more potential toxicity) since the SOC therapy is still effective and the difference seems to be not biologically relevant. Also puzzling is that the TPFR are cross-resistant to abema but MPFR is not, and then the difference between the double combination fulv+cdk4/6i and then triple combination is not significant in either MPFR or TPFR.

Figure 5.

the data does not fully support conclusions. The triple combination effect does not seem drastically different than double. This is similar to figure 2, but now in a model of fulv+cdk4/6i resistance versus figure 2 was only fulv resistant models. Also, they still did not address the differences of p-AKT levels between panel 5D and 5E even though they say it is the same.

Panel C: The levels of cleaved-PARP in TPFR cells are not different compared to vehicle or fulvestrant alone, which again disagrees with what the authors claim.

Panel D: No evidence that phospho Rb levels were assessed. Also the differences between the other proteins are very subtle and need detailed densitometry to address the issue.

In vivo data (Figures 3 and 6):

Figure 3. the data does not fully support conclusion. We had asked previously about the triple combination being significantly different from double but they claim that since they started with the larger tumor volume in Figure 3E supports the claim of triple being the better therapy. Lack of significant differences in panel D, diminish the enthusiasm for data in E

Raw data used for the statistical analysis should be included for panel E.

Panel F-authors do not address why the tumor volume was evaluated at 2 weeks instead at the end-point

Supplementary fig 4: Whenever combination index is evaluated treatment responses with the single agents should also be included in order to be able to adequately address the power of the combination treatment.

Panel H also need to show the staining of Ki67 and CC3 for the controls.

Figure 6:

Their data in this figure is contradictory to that shown in figure 4 that the difference between the growth of MPFR cells treated with the double combination and treated with the vehicle is not statistically significant No assessment of controls in their PDX models. Not clear why the tumor growth curve for the triple combination treatment (Fulv+CDK4/6i+AKTi) is not included

3. Clinical data:

Figure 7. it would have been important for them to show how the AKT inhibitor compares to mTOR or PI3K which are already being used in the clinic .

Reviewer #4:

Remarks to the Author:

Nice manuscript that highlights the role of AKT inhibition in HR+ breast cancer. Please consider the following comments:

1. Patients do not progress or respond. It is the disease that progresses or responds. Please modify language accordingly. I would urge authors to review the ASCO POST article (see Table in the article):

<https://ascopost.com/issues/april-10-2020/using-respectful-language-to-reduce-unconscious-bias-in-oncology-care/>

2. Authors make a case as to how mTOR inhibition leads to compensatory AKT activation. However, the authors do not provide any evidence of role of AKT inhibition in this setting, i.e cells lines/PDX models that are resistant to mTOR inhibitors (such as everolimus) and demonstrate activity of AKT inhibitor (such as capivasertib) with/without CDK 4/6 inhibitor in this setting. Similarly, how does mTOR inhibition compare to AKT inhibition. This is particularly relevant given clinical trials evaluating triplet therapy with endocrine therapy + CDK 4/6 inhibitor + mTOR inhibitor such as TRINITI trial have demonstrated efficacy. Impressive data with gedatolisib were also presented at AACR 2021. mTORi vs AKTi is important point and should be discussed.

3. Authors highlight that tumors with high expression of p-AKT correlates with shorter PFS, but pre-clinical data is missing. In other words, how PDX models with high (vs low) p-AKT expression respond to AKTi with/without CDK 4/6i.

4. The authors present clinical data related to high (vs low) expression of p-AKT. Since tumor genotyping is routinely done in the metastatic breast cancer setting, it might be helpful to provide the AKT mutation results (present vs absent) and association with p-AKT levels and clinical outcomes.

5. The authors provide COX proportional hazard ratio evaluating association between p-AKT and clinical outcomes. However, since multiple prognostic factors such as age (premenopausal vs postmenopausal), visceral mets (vs bone mets), no of prior lines, can impact progression-free survival, it would be important to adjust for these variables in multivariate regression models, otherwise results could be misleading.

Response to comments by Reviewer #4:

1. Patients do not progress or respond. It is the disease that progresses or responds. Please modify language accordingly. I would urge authors to review the ASCO POST article (see Table in the article):

<https://ascopost.com/issues/april-10-2020/using-respectful-language-to-reduce-unconscious-bias-in-oncology-care/>

Our response: As requested by the reviewer, we have modified the language according to the table presented in the ASCO POST article “Using Respectful Language to Reduce Unconscious Bias in Oncology Care.”

2. Authors make a case as to how mTOR inhibition leads to compensatory AKT activation. However, the authors do not provide any evidence of role of AKT inhibition in this setting, i.e cells lines/PDX models that are resistant to mTOR inhibitors (such as everolimus) and demonstrate activity of AKT inhibitor (such as capivasertib) with/without CDK 4/6 inhibitor in this setting. Similarly, how does mTOR inhibition compare to AKT inhibition. This is particularly relevant given clinical trials evaluating triplet therapy with endocrine therapy + CDK 4/6 inhibitor + mTOR inhibitor such as TRINITY trial have demonstrated efficacy. Impressive data with gedatolisib were also presented at AACR 2021. mTORi vs AKTi is important point and should be discussed.

Our response: As requested by the reviewer, we have performed additional experiments comparing the dual mTOR and PI3K inhibitor gedatolisib with AKT inhibitor in HR+ breast cancer cell lines resistant to combined CDK4/6i and fulvestrant (MPF-R and TPF-R). Triple combination with dual PI3K/mTORi induced a marked growth inhibition in MPF-R and TPF-R cells compared to standard combined CDK4/6i and fulvestrant. The growth inhibition induced in MPF-R and TPF-R cells by the triple combination with dual PI3K/mTORi was similar to that of triple combination with AKTi, suggesting that these drug combinations have comparable efficacies. These data are included in Supplementary Figure S6, Results (p 12) and Discussion (p 20).

3. Authors highlight that tumors with high expression of p-AKT correlates with shorter PFS, but pre-clinical data is missing. In other words, how PDX models with high (vs low) p-AKT expression respond to AKTi with/without CDK 4/6i.

Our response: We agree with the reviewer that evaluation of response to AKTi with/without CDK4/6i in PDX models with high vs. low p-AKT expression would be relevant. However, it is extremely challenging to develop ER+ PDXs resistant to combined CDK4/6i and endocrine therapy with variable levels of p-AKT S473. The PDX models currently available with variable levels of p-AKT are not resistant to CDK4/6i and/or endocrine therapy (Gris-Oliver et al, Clin Cancer Res., 2020) and thus are not relevant to include in our study. Nevertheless, based on our data from in vitro and cell line xenograft models showing high p-AKT in CDK4/6i and fulvestrant-resistant tumors, which respond to triple combination with AKTi, and the data published by Gris-Oliver et al, 2020 showing that PDXs exhibiting high p-AKT levels correlate with higher sensitivity to AKTi, we believe we have provided strong support for the addition of AKTi in ER+ breast cancer patients who progress on CDK4/6i and endocrine therapy.

4. The authors present clinical data related to high (vs low) expression of p-AKT. Since tumor genotyping is routinely done in the metastatic breast cancer setting, it might be helpful to provide the AKT mutation results (present vs absent) and association with p-AKT levels and clinical

outcomes.

Our response: We agree with the reviewer that it would have been interesting to provide AKT mutation status of the clinical samples, but genotyping of metastatic breast cancers has just recently become available at our hospital and only on selected patients. Thus, AKT mutation status is not available for this cohort of patients treated with combined CDK4/6i and endocrine therapy. However, since AKT mutations are only present in 3-6% (Xiao et al, J. Cancer, 2021, Razavi et al, Cancer Cell, 2018) of ER+ breast tumors and the size of our cohort is limited, it is not likely that we would be able to demonstrate a significant association between AKT mutation status and p-AKT levels and clinical outcomes.

5. The authors provide COX proportional hazard ratio evaluating association between p-AKT and clinical outcomes. However, since multiple prognostic factors such as age (premenopausal vs postmenopausal), visceral mets (vs bone mets), no of prior lines, can impact progression-free survival, it would be important to adjust for these variables in multivariate regression models, otherwise results could be misleading.

Our response: We agree with the reviewer that it would be important to adjust for the proposed variables using multivariate regression models. However, the effective sample size (no. of death/relapse events) in the p-AKT low and p-AKT high is 33 and 10, respectively, which limits the number of variables that can be included in the analysis. Thus, the sample size in the study is not sufficient to adjust for these variables. Among the clinical parameters suggested to be included by the reviewer, only the no. of prior lines showed to be significant in the univariate analysis (Supplementary Table S2). Although age and site of relapse would also be expected to be significant, as these parameters are known to be associated with disease survival, we do not have sufficient power to show this correlation and to adjust for these parameters. Nevertheless, evaluation of the distribution of the proposed variables between the groups shows that age, no. of metastasis, time to recurrence, line of therapy and site of relapse are evenly distributed between p-AKT low and high groups. The even distribution of these variables between the two groups might indicate that high p-AKT would indeed be statistically significant in a larger population of patients after adjusting for the clinical variables. Although a comprehensive analysis of the clinical significance of p-AKT is not possible in our study, we believe that our data clearly indicate that the level of p-AKT is associated with CDK4/6 resistance and provide a firm basis and an interesting perspective for future studies. These data are included in Supplementary Tables S2 and S3, Results (p 17) and Discussion (p 21).

Response to comments by Reviewer #3:

In vitro data (Figures 1, 2, 4 and 5)

Figure 1. The differences the authors show while statistically significant are unlikely to be biologically significant. The reasons are as follows: taking in consideration the results for the growth assay, cell viability, and the long-term growth assay, only one out of 3 models (182R-1) showed an evident effect. A second model (ZR-75-1R) showed no differences comparing controls to double combination in 2 out of 3 assays, specifically in the long-term growth assay, except for vehicle, all treatments were equally effective. The third model, T47DR, although the differences

were statistically significant, the effect seems not very clear in the growth assay and long-term growth assay, since in panel P, T47DR cells are still responsive to fulv+cdk4/6i.

Our response: We appreciate that the reviewer now agrees that the differences observed are statistically significant. We respectfully disagree that the findings are unlikely to be of biological significance, although a clinical trial will ultimately determine this. Concerning the third fulvestrant-resistant model T47DR, we are puzzled by the reviewer's claim that the effect seems not very clear in the growth assay and long-term growth assay. Our data clearly shows a very significant difference in growth of T47R cells treated with triple combination compared to standard combined CDK4/6i and fulvestrant ($p < 0.0001$, Fig. 1F). Importantly, we observe in the long-term experiment that T47DR resistant clones start to develop at week 5 when cells are treated with fulvestrant + CDK4/6i, while resistant clones do not develop during the whole 12 weeks of the experiment when treating T47DR cells with the triple combination. As T47DR is a fulvestrant-resistant model, it is expected that these cells show some response to fulvestrant + CDK4/6i treatment in the short- and long-term growth assay.

Combination index: Authors included new data with the other models, however, the main concern remains, and it is even more evident since in all the 3 models the double and triple combination as well as the other combinations tested showed to be synergistic. Although, as the authors discussed, this does not suggest that treatments are equally affective, these new results reduce the enthusiasm for the triple combination and raise the question about the advantage to add an additional therapy, which could cause additional adverse events.

Our response: We appreciate that the reviewer now agrees that although several combination treatments exhibit synergism this does not suggest that treatments are equally effective. It is not clear why our new results reduce the reviewer's enthusiasm for the triple combination as this is not specified.

Normalization: To properly normalize the data, the authors need to use the same Y axis from model to model, not different Y axis to amplify their effects. Additional dose ranges to use: Not clear why the authors did not compare the more physiological dose of palbo in the ZR751 line.

Our response: We have used different Y axis in the graphs to properly show the results and not to amplify the effect, as suggested by the reviewer. The different range in the Y axis is a result of the expected biological variation between cell models. The important point of these graphs is to compare the effect of the different treatments within each model rather than comparing the treatments between the different models.

The palbociclib dose selected in ZR-75-1 cells was based on the dose response curve and calculated IC50. Based on these results, ZR-75-1 cells seem to be less sensitive to palbociclib than the cells from other models, and therefore a higher concentration of palbociclib (5 μ M) was used for ZR-75-1 cells. This concentration is still within the physiologically relevant concentration (5 vs 3.88 μ M). We do not find the lower sensitivity of ZR-75-1 cells to palbociclib of concern for the overall conclusion of the study, and we believe this is a result of the normal biological variation observed between different ER+ breast tumors.

Uniform characterization of the cell lines: The authors did not characterize all the cell lines used in a uniform fashion. They needed to show the protein expression of all the pathways studied (additional to the ones showed in fig 2), not just the p-Rb/Rb pathway. It seems that the authors are

cherry picking which proteins to examine in what cell lines and for what combination. For example, since the authors are proposing triple combination, it would be good to also CDK4, CDK6, Cyclin D, etc;

Our response: It seems that the reviewer has overlooked the fact that we have also examined the ER and AKT pathways in addition to the p-Rb/Rb pathway in Figures 2 and 5. These are the central pathways inhibited in this study and these analyses were performed uniformly across all 3 fulvestrant-resistant cell lines (Figure 2E), as well as the combined fulvestrant+CDK4/6i-resistant cell lines (Figure 5D). The effect of the different inhibitors should be evaluated by examining the expression of the downstream proteins and not the specific targeted proteins, and thus evaluation of CDK4, CDK6 and Cyclin D does not seem relevant.

Results are not consistent with the claims: Authors state on page 8 line 174-175 that high levels of p-akt correlates with high sensitivity to AKT inhibitor; however, there is no differences between double combo fulv+cdk4-6 and the triple combination. Again, only 2 models (182R1 and ZR751R) show that p-AKTS473 increase after triple combination compared to fulv+cdk4-6, but this is not in agreement with the results in figure 1. Only the result for 182R1 correlates (high p-akt, high sensitivity to triple combo). ZR751R show high p-akt but was not sensitive to the triple combination.

Our response: Assessing the correlation between p-AKT levels and sensitivity to AKTi requires examination of p-AKT baseline expression in untreated cells (Fig. 2F). MCF-7 cells exhibited the lowest p-AKT S473 across all models and these cells were treated with the highest dose of AKTi (500nM), while ZR-75-1 and T47D cells exhibited higher p-AKT S473 and were treated with lower doses of AKTi (150 nM and 200 nM, respectively). These data suggest that high p-AKT S473 levels correlate with higher AKTi sensitivity, which is also supported by the findings of Gris-Oliver et al. (Clin. Cancer Res. **26**, 3720-3731, 2020).

Additional minor issues with Figure 2 are as follows

o No densitometry for panel E protein.

Our response: In the last revision of the manuscript, we included the densitometry of proteins in panel F, which we believe are the most relevant proteins to quantify.

o Inconsistencies between panels E and F: p-aktS473 is high for 182R1 cells in panel F but very low in panel E (vehicle), same thing for T47DR, ZR751 and ZR751R

Our response: Blots in panel F were exposed for longer time than blots in panel E, which explains the differences in protein band intensity observed between these 2 panels.

Figure 4.

Panel F-it is curious that the differences between black and blue lines is not statistically significant, but the differences between blue and red (which is similar to the black and blue, around 2 times) is statistically significant. A more rigorous review of the statistical analysis is required.

Our response: We have repeated the statistical analysis for the data in Figure 4F and obtained the same results.

The outgrowth of TPFR resistant colonies treated with CDK4/6i start the log phase after week 7, which suggest that these cells are still responsive and are not fully resistant as the authors state. It is an important point because, again, it raises the question about the advantage to add an additional therapy (more potential toxicity) since the SOC therapy is still effective and the difference seems to be not biologically relevant. Also puzzling is that the TPFR are cross-resistant to abema but MPFR is not, and then the difference between the double combination fulv+cdk4/6i and then triple combination is not significant in either MPFR or TPFR.

Our response: The outgrowth of TPFR-resistant colonies treated with CDK4/6i + Fulv starts as early as week 3, which does not suggest that these cells are still responsive to this treatment. We respectfully disagree with the reviewer that the difference between the triple combination (where no resistant colonies were observed during the whole 12 weeks of the experiment) and the CDK4/6i + fulv treatment (where resistant colonies developed at week 3) is not biologically relevant.

Figure 5.

the data does not fully support conclusions. The triple combination effect does not seem drastically different than double. This is similar to figure 2, but now in a model of fulv+cdk4/6i resistance versus figure 2 was only fulv resistant models. Also, they still did not address the differences of p-AKT levels between panel 5D and 5E even though they say it is the same.

Our response: It is not clear what data set in figures 2 and 5 the reviewer refers to when stating that the effect of the triple combination does not seem drastically different to that of the double. There is a clear decrease in p-PRAS40 and p-S6 levels in all 5 resistant cells treated with triple combination compared to standard combined CDK4/6i+fulvestrant (Figs. 2E and 5D). Furthermore, ER and p-Rb expression are also reduced in most of the resistant cells after treatment with triple combination compared to CDK4/6i+fulvestrant (Figs. 2E and 5D). The Western blot in panel 5E has been exposed longer than 5D, which explains the differences in p-AKT levels in M-S and MPF-R cells between the two panels. The higher level of p-AKT in TPF-R vs. T-S cells in 5D is due to higher loading protein (higher GAPDH levels) in TPF-R compared to T-S (Supplementary Fig. S11).

Panel C: The levels of cleaved-PARP in TPFR cells are not different compared to vehicle or fulvestrant alone, which again disagrees with what the authors claim.

Our response: We believe the reviewer may have overlooked the statement on page 13 of Results “Furthermore, we observed a marked increase in apoptosis and cleaved PARP levels in MPF-R cells treated with the triple combination compared to the standard fulvestrant and CDK4/6i combination, although these changes were not observed in TPF-R cells (Fig. 5A-C).” More importantly, we observe a marked reduction of p-PRAS40 and p-S6 levels in TPF-R cells treated with triple combination compared to combined CDK4/6i and fulvestrant, which concurs with our findings in short- and long-term growth and viability assays (Fig 4F, 4H and 4J).

Panel D: No evidence that phospho Rb levels were assessed. Also the differences between the other proteins are very subtle and need detailed densitometry to address the issue.

Our response: Phospho Rb levels were assessed and shown in Supplementary Fig. S7 as indicated in Results (p 13). We respectfully disagree with the reviewer that the differences between the other proteins are very subtle.

In vivo data (Figures 3 and 6):

Figure 3. the data does not fully support conclusion. We had asked previously about the triple combination being significantly different from double but they claim that since they started with the larger tumor volume in Figure 3E supports the claim of triple being the better therapy.

Lack of significant differences in panel D, diminish the enthusiasm for data in E

Our response: As explained in the last revision of the manuscript, both the dual and triple drug combinations completely suppressed outgrowth of the tumors when they are small at treatment initiation (panel D). In contrast, when the tumors were larger at treatment initiation, the triple combination induced a statistically significant growth inhibition compared to the combined fulv+CDK4/6i, as analyzed by two-tailed t-test at the endpoint ($p < 0.05$) (panel E). We have also included the non-parametric Wilcoxon test showing a significant difference in tumor volume between the two treatment groups ($p = 0.009$, Fig. 3E). Further, we also observed statistically significant differences in the growth rates between the two treatment arms by using linear mixed effects models (GR 7.47, $p = 0.009$, CI 1.88-13.07). Therefore, we respectfully disagree with the reviewer that “the data does not fully support conclusion”.

Raw data used for the statistical analysis should be included for panel E.

Our response: As requested by the reviewer, we have included a file for the reviewer with the raw data used for the statistical analysis of panel E. All raw data will be made available in a data repository when the manuscript is accepted for publication.

Panel F-authors do not address why the tumor volume was evaluated at 2 weeks instead at the endpoint

Our response: There is not a panel 3F in this version. The reviewer may be referring to Supplementary Fig. S4E and S4F where the tumor volume was evaluated at the endpoint, which in this experiment was 2 weeks.

Supplementary fig 4: Whenever combination index is evaluated treatment responses with the single agents should also be included in order to be able to adequately address the power of the combination treatment.

Our response: The data from combination index evaluation is not included in Supplementary Fig. S4 but in Supplementary Fig. S2. Treatment responses with the single agents for each cell line are already included in Supplementary Fig. S1.

Panel H also need to show the staining of Ki67 and CC3 for the controls.

Our response: Panel H in Supplementary Fig. S4 refers to data from the experiment in Fig. 3E where no controls were included. Controls were included in Supplementary Fig. S4G, which refers to data from the experiment in Fig. 3D, which is a similar experiment to Fig. 3E but with smaller tumors, where controls showed no effect on tumor growth. For this reason, no controls were included in the experiment in Fig. 3E and Supplementary Fig. S4H.

Figure 6:

Their data in this figure is contradictory to that shown in figure 4 that the difference between the growth of MPFR cells treated with the double combination and treated with the vehicle is not statistically significant. No assessment of controls in their PDX models. Not clear why the tumor growth curve for the triple combination treatment (Fulv+CDK4/6i+AKTi) is not included

Our response: It is not surprising that cell lines in general, particularly in this case MPF-R cells, behave differently when growing in vitro and in vivo. In the mice experiments with MPF-R cells, treatment was only initiated one month after cells were injected orthotopically in mice, which could explain why MPF-R cells became more sensitive to combined CDK4/6i and fulvestrant in vivo. We are puzzled by the reviewer's claims that there is no assessment of controls in the PDX models as panel C clearly shows that the sensitive untreated PDXs exhibited a growth curve comparable to that of the PDXs resistant and treated with CDK4/6i+fulvestrant and the sensitive treated PDXs growth is completely inhibited by CDK4/6i+fulvestrant.

3. Clinical data:

Figure 7. it would have been important for them to show how the AKT inhibitor compares to mTOR or PI3K which are already being used in the clinic.

Our response: As described above in our response to Reviewer's #4 comment 2, we evaluated the efficacy of the dual mTOR and PI3K inhibitor gedatolisib in combination with CDK4/6i and fulvestrant and compared these results with the triple combination with AKT inhibitor in HR+ breast cancer cell lines resistant to combined CDK4/6i and fulvestrant (MPF-R and TPF-R). Triple combination with dual PI3K/mTORi showed a marked growth inhibition in MPF-R and TPF-R cells compared to standard combined CDK4/6i and fulvestrant. The growth inhibition induced in MPF-R and TPF-R cells by the triple combination with dual PI3K/mTORi was similar to that of the triple combination with AKTi, suggesting that these drug combinations have comparable efficacies. These data are included in Supplementary Figure S6, Results (p 12) and Discussion (p 20).

Reviewers' Comments:

Reviewer #4:

Remarks to the Author:

Thank you for addressing the comments. The authors kindly acknowledged the two comments (#4,5) that could not be addressed completely due to small sample size (multivariate analysis) and lack of availability of data (AKT genotyping results). This point could be mentioned as a statement in limitations section.

Reviewer #4 (Remarks to the Author):

Thank you for addressing the comments. The authors kindly acknowledged the two comments (#4,5) that could not be addressed completely due to small sample size (multivariate analysis) and lack of availability of data (AKT genotyping results). This point could be mentioned as a statement in limitations section.

Our Response: As requested by the reviewer, we have included a short paragraph in the Discussion highlighting these limitations (page 22).